# Temporal Conditioning Spiking Latent Variable Models of the Neural Response to Natural Visual Scenes

**Gehua Ma**
College of Computer Science and Technology
Zhejiang University
gehuama@icloud.com

**Runhao Jiang**
College of Computer Science and Technology
Zhejiang University
RhJiang@zju.edu.cn

**Rui Yan**
College of Computer Science and Technology
Zhejiang University of Technology
ryan@zjut.edu.cn

**Huajin Tang**[*]
College of Computer Science and Technology
Zhejiang University

## Abstract

Developing computational models of neural response is crucial for understanding sensory processing and neural computations. Current state-of-the-art neural network methods use temporal filters to handle temporal dependencies, resulting in an unrealistic and inflexible processing paradigm. Meanwhile, these methods target trial-averaged firing rates and fail to capture important features in spike trains. This work presents the temporal conditioning spiking latent variable models (TeCoS-LVM) to simulate the neural response to natural visual stimuli. We use spiking neurons to produce spike outputs that directly match the recorded trains. This approach helps to avoid losing information embedded in the original spike trains. We exclude the temporal dimension from the model parameter space and introduce a temporal conditioning operation to allow the model to adaptively explore and exploit temporal dependencies in stimuli sequences in a *natural paradigm*. We show that TeCoS-LVM models can produce more realistic spike activities and accurately fit spike statistics than powerful alternatives. Additionally, learned TeCoS-LVM models can generalize well to longer time scales. Overall, while remaining computationally tractable, our model effectively captures key features of neural coding systems. It thus provides a useful tool for building accurate predictive computational accounts for various sensory perception circuits.

## 1  Introduction

Building precise computational models of neural response to natural visual stimuli is a fundamental scientific problem in sensory neuroscience. These models can offer insights into neural circuit computations, reveal new mechanisms, and validate theoretical predictions [1, 2, 3, 4, 5, 6]. However, constructing such models is challenging due to the complex nonlinear processes involved in neural coding, such as synaptic transmission and spiking dynamics. For modeling retinal responses, early attempts using linear-nonlinear (LN) models and generalized linear models (GLMs) were successful with simple data such as white noise [2, 7] but fell short with more complex stimuli like natural visual scenes [8, 9]. Artificial neural networks (ANNs), which are powerful function approximators and loosely resemble biological neurons and architectures [10, 11], have shown promise in modeling

---

[*]Correspondence: htang@zju.edu.cn

37th Conference on Neural Information Processing Systems (NeurIPS 2023).

the visual stimuli coding process through various ANN-based methods [9, 12, 13, 14, 15]. Recent research has demonstrated that complex neural activity can be well represented in a low-dimensional space [16, 17]. This has led to growing interest in a type of neural network known as latent variable models (LVMs). LVMs strike a balance between model accuracy and representation space simplicity, enabling accurate neural coding modeling and interpretation of neural activity in a low-dimensional space [18, 19, 20, 21, 22, 23, 24]. Although the current state-of-the-art neural network models for visual neural coding have produced decent results and provided some exciting insights, they have two major limitations:

- Most of these works focus on simulating the firing rates of real neurons. This is a decent choice (following the classic Poisson LN/GLM models) for artificial neuron networks, which are essentially real-valued processing-based. However, as a trial-averaged spike statistic, firing rates only characterize some aspects of the original spike train [25]. As a result, directly using firing rates as the target may result in losing information embedded in the original spike trains [26, 27].
- Existing models mostly employ fixed-length temporal filters. For example, the CNN approach [9] concatenates stimuli within a fixed duration as input and processes these inputs using temporal filters. Therefore, they cannot process long stimuli sequences like a real neural circuit. Instead, they must slice the long sequences into fixed-length short segments and process them separately, thus losing biological realism (see also Fig. 2A). Secondly, in simulation, the learned models can only take inputs of the same length as during the training phase, thus limiting their flexibility.

Although these two points are important for a realistic computational model, an approach that addresses these limitations is still lacking. This work introduces the TeCoS-LVM (temporal conditioning spiking LVM) models of the neural responses to natural visual stimuli. We employ spiking neurons to allow the model to aim directly at producing realistic spike trains. This avoids spike train information loss that might occur when targeting spike statistics. To address the second limitation, we completely exclude the temporal dimension from the parameter space and introduce a temporal conditioning operation to handle the temporal dependencies. Inspired by the information compression in biological coding systems, we formalize TeCoS-LVM models within the information bottleneck framework of LVMs and introduce a general learning method for them. Evaluations on real neural recordings demonstrate that TeCoS-LVM models accurately fit spike statistics and produce realistic spike activities. Further simulations show that TeCoS-LVM models learned on short sequences can generalize well to longer time scales and exhibit memory mechanisms similar to those in biological cognitive circuits.

## 2 Preliminaries

**Leaky integrate-and-fire spiking neuron** We adopt the Leaky Integrate-and-Fire (LIF) neuron model in this work, which briefly describes the sub-threshold membrane potential dynamics as $\tau_m \frac{du}{dt} = -(u - u_{\text{reset}}) + RI(t)$, where $u_t$ denotes membrane potential, $R, \tau_m$ are membrane resistance, time constant, and $I$ is the input current. $v_{\text{th}}, u_{\text{reset}}$ denote the firing threshold, resting potential, respectively. In practice, it leads to the following discrete-time computational form [28, 29, 30]. The sub-threshold dynamic, firing, and resetting are written as $u_t = \tau u_{t-1} + I_t$; $o_t = \text{Heaviside}(u_t - v_{\text{th}})$; $u_t = u_{\text{reset}}$ if $o_t = 1$, where $o_t$ is the spike output, $I_t$ is the input current, usually transformed by a parameterized mapping, and $\tau$ is the membrane time constant. In this discrete form, the membrane constant $\tau_m$ is combined with timestep $dt$ for simplification, as in all our experiments, the length of simulation timestep $dt$ is fixed. To introduce a simple model of neuronal spiking and refractoriness, we assume $v_{\text{th}} = 1, \tau = 0.5$ is a fixed constant, and $u_{\text{reset}} = 0$ for all spiking neurons throughout this research.

**Variational information bottleneck** The information bottleneck (IB) principle [31] offers an appealing framework to formulate LVMs. In short, a desired LVM should have latent representations that are maximally expressive regarding its target while being maximally compressive about its input. Let us consider a latent variable model $\theta$ with input $\mathbf{x}$, target $\mathbf{y}$, and latent representation $\mathbf{z}$ defined by a parametric encoder $q(\mathbf{z}|\mathbf{x}; \theta)$. Let $I(\mathbf{z}, \mathbf{y}; \theta)$ be the mutual information between the $\mathbf{z}$ and $\mathbf{y}$, and $I(\mathbf{z}, \mathbf{x}; \theta)$ be the mutual information between $\mathbf{z}$ and $\mathbf{x}$, IB suggests an objective

$$\max_{\theta} I(\mathbf{z}, \mathbf{y}; \theta) \text{ s.t. } I(\mathbf{z}, \mathbf{x}; \theta) < I_c, \tag{1}$$

where $I_c$ is the information constraint. With the introduction of a Lagrange multiplier $\beta$ [31], the IB objective is equivalent to

$$\max_{\theta} [I(\mathbf{z}, \mathbf{y}; \theta) - \beta I(\mathbf{z}, \mathbf{x}; \theta)]. \tag{2}$$

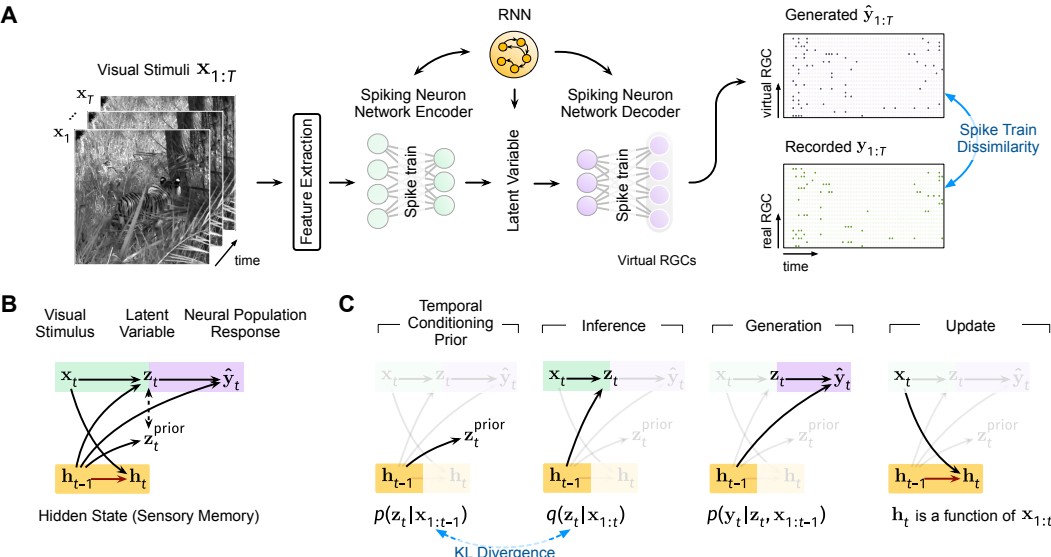

Figure 1: Overview of our approach. **A.** Graphical illustration of TeCoS-LVM models of retinal neural response to natural visual scenes. Our model can directly generate neural response sequences in real-time, thus faithfully simulating the real neural computation process (see also Fig. 2A). **B.** Full graphical illustration of the computational operations. **C.** Separate illustrations of the prior, encoder inference, decoder generation, and hidden state update operations. With the introduction of the hidden state (acts as the sensory memory), our prior, encoder and decoder are linked to the entire stimuli sequence $\mathbf{x}_{1:t}$ rather than just the current stimulus $\mathbf{x}_t$ (see also Appendix 3.1). Hence, although we completely exclude the time dimension from the model parameter space, TeCoS-LVM can still adaptively explore and exploit temporal information for predictive (generative) modeling.

Deep variational IB [32, 33] leverages variational inference to construct a lower bound on the IB objective in (2). By assuming the factorization $p(\mathbf{x}, \mathbf{y}, \mathbf{z}) = p(\mathbf{z}|\mathbf{x})p(\mathbf{y}|\mathbf{x})p(\mathbf{x})$, we have an equivalent objective that comprises one predictive term and one compressive term (refer to Appendix D) as follows,

$$\min_{\boldsymbol{\theta}} \quad \underbrace{\mathbb{E}_{q(\mathbf{z}|\mathbf{x};\boldsymbol{\theta})}[-\log p(\mathbf{y}|\mathbf{z};\boldsymbol{\theta})]}_{\mathcal{L}^{\text{pred}}:\text{ encouraging predictive power}} + \beta \cdot \underbrace{\text{KL}\left[q(\mathbf{z}|\mathbf{x};\boldsymbol{\theta})\|p(\mathbf{z})\right]}_{\mathcal{L}^{\text{comp}}:\text{ encouraging compression}} , \qquad (3)$$

where $\text{KL}[Q\|P]$ is the Kullback-Leibler divergence between two distributions, and $p(\mathbf{y}|\mathbf{z};\theta)$ is a parameterized decoder.

## 3 Methodologies

### 3.1 Temporal Conditioning Spiking Latent Variable Model

**Basic formulation** (see also Appendix D) We denote a sequence of visual stimuli as $\mathbf{x} = (\mathbf{x}_1, \cdots, \mathbf{x}_t, \cdots, \mathbf{x}_T) \in \mathbb{R}^{T \times \dim[\mathbf{x}_t]}$, $\dim[\mathbf{x}_t]$ stands for the dimension of $\mathbf{x}_t$. At each timestep $t$, one high-dimensional visual stimulus $\mathbf{x}_t$ is received. We want to predict the corresponding neural population response $\mathbf{y}_t \in \{0, 1\}^{\dim[\mathbf{y}_t]}$, where $\dim[\mathbf{y}_t]$ denotes the number of retinal ganglion cells (RGCs). This is implemented by an LVM which first compresses the visual stimuli into a low-dimensional latent representation $\mathbf{z}_t \in \mathbb{R}^{\dim[\mathbf{z}_t]}$, and then decodes the neural population response from it. Biological neural coding can effectively compress observed stimuli while retaining the informative contents [34, 35, 36, 37]. Therefore, we further encourage this LVM to construct a latent space in which $\mathbf{z}$ have maximal predictive power regarding response $\mathbf{y}$ while being maximally compressive about input stimuli $\mathbf{x}$ [38]. As a result, our target to model the neural coding process of visual stimuli turns to an optimization problem of minimizing the loss function (3) presented in the variational IB framework.

However, assuming independence along the temporal dimension becomes inappropriate due to the complex temporal dependencies in stimuli-neural response modeling [39, 40, 41]. To address this, we introduce a hidden state into the prior, encoder, and decoder of the latent variable model. This hidden state maintains earlier stimuli information [42, 43], allowing the entire inference-generation

process to be conditioned on the entire stimuli sequence $\mathbf{x}_{1:t}$ rather than just $\mathbf{x}_t$ (Fig. 5). To enable adaptive exploitation and accumulation of temporal dependencies within the stimuli sequence, we employ a recurrent network to update the hidden state [44, 45, 46, 47], formally,

$$\mathbf{h}_t = f_{\text{RNN}}(\mathbf{x}_t, \mathbf{h}_{t-1}). \tag{4}$$

The hidden state here serves as the sensory memory [48], akin to the Tolman-Eichenbaum Machine [49], which employs an attractor network to store and retrieve memories.

Note that, as we focus on high-dimensional visual stimuli here, we use a spiking convolutional feature extractor to reduce the dimensionality of the stimuli. It is shared in the hidden state update and the temporal conditioning encoder inference process (as illustrated in Fig. 1). To make our notation simpler and easier to understand during reading, we do not explicitly denote the feature extractor in our formulations (stimuli features for de facto computation, but still marked as $\mathbf{x}$).

**Temporal conditioning prior**   With the introduction of the hidden state, the prior over latent variable is no longer a standard isotropic Gaussian. Instead, it becomes a parameterized conditional distribution that depends on the hidden state, which carries information about previous stimuli. Formally, the latent variable follows the distribution $p(\mathbf{z}_t|\mathbf{x}_{1:t-1})$, given by

$$p(\mathbf{z}_t|\mathbf{x}_{1:t-1}) = \mathcal{N}\Big(\mathbf{z}_t; \underbrace{\phi^{\text{prior}}(\mathbf{h}_{t-1})}_{\text{Mean } \mu}, \text{diag}\big(\underbrace{\phi^{\text{prior}}(\mathbf{h}_{t-1})}_{\text{Variance } \sigma^2}\big)\Big), \tag{5}$$

where $\phi^{\text{prior}}$ denotes the parametric model to compute the means and variances, a spiking MLP. In particular, the real-valued means and variations are calculated through the linear readout synapses of spiking neurons.

**Temporal conditioning encoder**   In a similar manner, the encoder is not only a function of $\mathbf{x}_t$, but also a function of $\mathbf{h}_{t-1}$. By Eq. 4, the hidden state $\mathbf{h}_{t-1}$ is a function of $\mathbf{x}_{1:t-1}$. Hence, the temporal conditioning encoder defines the distribution $q(\mathbf{z}_t|\mathbf{x}_{1:t})$. We can write

$$q(\mathbf{z}_t|\mathbf{x}_{1:t}) = \mathcal{N}\Big(\mathbf{z}_t; \psi^{\text{enc}}(\mathbf{x}_t, \mathbf{h}_{t-1}), \text{diag}(\psi^{\text{enc}}(\mathbf{x}_t, \mathbf{h}_{t-1}))\Big), \tag{6}$$

where $\psi^{\text{enc}}$ is the parameterized model for computing the variational posterior distribution, which is also a spiking MLP.

**Temporal conditioning decoder**   The decoder generates the neural population responses $\mathbf{y}_t$ only using the latent representation $\mathbf{z}_t$ and hidden state $\mathbf{h}_{t-1}$ as inputs. As the hidden state is a function of previous stimuli, the decoder defines the distribution $p(\mathbf{y}_t|\mathbf{z}_t, \mathbf{x}_{1:t-1}; \psi^{\text{dec}})$, where $\psi^{\text{dec}}$ stands for the decoder parameters. Since we use spiking neurons, the decoder will directly output spike trains that simulate the recorded neural population responses.

**Model learning**   The TeCoS-LVM models are optimized to minimize a loss function (Eq. 3, see also Appendix D) that has the form:

$$\mathcal{L} = \mathcal{L}^{\text{pred}} + \beta \mathcal{L}^{\text{comp}}, \tag{7}$$

and all synaptic weights are optimized jointly during the learning. To elucidate the loss function, we first examine the compressive term. As we adopt Gaussian distributions in our temporal conditioning prior and encoder, the compressive term composed of KL divergences can be calculated analytically. In particular, we have,

$$\mathcal{L}^{\text{comp}} = \frac{1}{T} \sum_{t=1}^{T} \text{KL}[q(\mathbf{z}_t|\mathbf{x}_{1:t}; \psi^{\text{enc}}) \| p(\mathbf{z}_t|\mathbf{x}_{1:t-1}; \phi^{\text{prior}})]. \tag{8}$$

On the other hand, since our model directly outputs spike trains, we adopt the Maximum Mean Discrepancy (MMD) as the predictive loss term [26, 50]. Also, we use the first-order postsynaptic potential (PSP) kernel that can effectively depict the temporal dependencies in spike train data [51, 52]. Denoting the predicted, recorded spike trains as $\hat{\mathbf{y}}$, $\mathbf{y}$, respectively, we can write the PSP kernel MMD predictive loss as

$$\mathcal{L}^{\text{pred}} = \frac{1}{T} \sum_{t=1}^{T} \sum_{\tau=1}^{t} \left\| \text{PSP}(\hat{\mathbf{y}}_{1:\tau}) - \text{PSP}(\mathbf{y}_{1:\tau}) \right\|^2, \tag{9}$$

where $\text{PSP}(\mathbf{y}_{1:\tau}) = (1 - \frac{1}{\tau_s})\text{PSP}(\mathbf{y}_{1:\tau-1}) + \frac{1}{\tau_s}\mathbf{y}_\tau$, and we set the time constant $\tau_s = 2$.

## 3.2 TeCoS-LVM Models

We shall consider two types of TeCoS-LVM models, `TeCoS-LVM` and `TeCoS-LVM Noisy` in this work. Specifically:

- In `TeCoS-LVM`, all spiking neurons are LIF neurons.
- In `TeCoS-LVM Noisy`, all spiking neurons are Noisy LIF spiking neurons ([53], refer to Appendix C for details). Using Noisy LIF neuron allows TeCoS-LVM to have neuron-level stochasticity, which is considered a crucial component in biological neural computation [54, 55, 56].

# 4 Experiments

## 4.1 Dataset and Baselines

We perform evaluations and analyses on real neural recordings from RGCs of dark-adapted axolotl salamander retinas [57]. The dataset contains spike responses of two retinas on two movies (see also Appendix E). We partitioned all records into stimuli-response sample pairs of 1 second (30 time bins) and down-sampled the frames to 90 pixel×90 pixel. This results in four datasets, each split into non-overlapping train/test (50%/50%) parts. We shall refer to these four datasets as follows for brevity. **Movie 1 Retina 1**: records of 38 RGCs on movie 1 ("salamander movie"), 75 repetitions. **Movie 1 Retina 2**: records of 49 RGCs on movie 1, 30 repetitions. **Movie 2 Retina 1**: records of 38 RGCs on movie 2 ("wildlife movie"), 107 repetitions. **Movie 2 Retina 2**: records of 49 RGCs on movie 2, 42 repetitions.

**CNN model**  We use the state-of-the-art CNN model [9, 15] for comparison. In the CNN model, the network's predicted outputs are the average maximum likelihood estimation of retina responses in firing rates based on spatiotemporal (frames are concatenated on the channel dimension) input (see also Appendix G).

**IB-Disjoint**  IB-Disjoint [21] is a high-performance model that performs similarly to the IB-GP model that uses a Gaussian Process prior [21]. This model employs an isotropic Gaussian as the prior over latents. It is similar to the vanilla variational IB, where the latent variables are assumed to be independent along the temporal dimension.

## 4.2 Metrics and Features for Evaluations and Visualizations

**Pearson correlation coefficient**  This metric evaluates the model performance by calculating the Pearson correlation coefficient between the recorded and predicted firing rates [9, 14, 15]. The higher the value, the better the performance. Refer to Appendix F for more details.

**Spike train dissimilarity**  This metric assesses the model performance by computing the dissimilarity between recorded and predicted spike trains. A lower value indicates better model performance. Here we use the MMD with a first-order PSP kernel [51, 52] to measure the spike train dissimilarity [27, 50, 58] (see also Appendix F).

**Spike autocorrelogram**  The spike autocorrelogram is computed by counting the number of spikes that occur around each spike within a predefined time window [14, 9]. The resulting trace is then normalized to its maximum value (which occurs at the origin of the time axis by construction), and the maximum value is set to zero for better visualization. Refer to Appendix F for more details.

## 4.3 Experimental Details

TeCoS-LVM hyper-parameters were fixed to be the same on all four datasets. We set the latent variable dimension to 32 and the hidden state dimension to 64 by default (see also Appendix A). We used the Adam optimizer ($\beta_1 = 0.9, \beta_2 = 0.999$) with a cosine-decay learning rate scheduler [59], starting at a rate of 0.0003. The mini-batch size was set to 64, and the models were trained for 64 epochs. We used the same architectures to implement TeCoS-LVM models. The factor $\beta$ in the loss function (7) is set to 0.01 by default to balance information compression and predictive power [32, 60]. More experimental details are presented in Appendix G.

At test time, the test samples have the same length as the training samples (30 bins, 1 second) by default. However, the learned model can handle longer test samples as our model excludes the time dimension from its parameter space. This will be discussed further in the Results sub-section. We tested TeCoS-LVM models with a "warmup period" of 0.5 seconds (15 bins), during which the

model's predictions will be discarded (see also Fig. 4A). This was done because our hidden state was zero-initialized, making the early predictions less accurate.

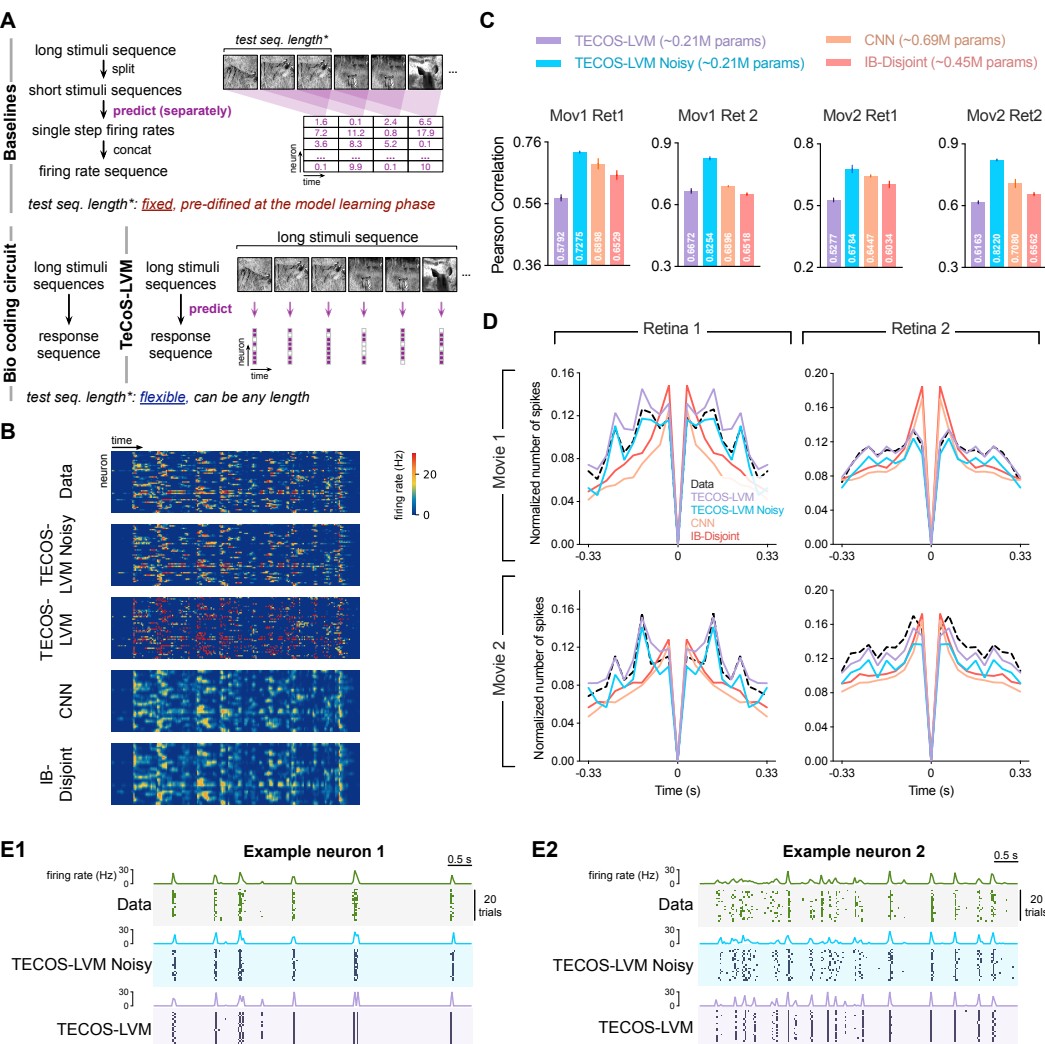

Figure 2: TeCoS-LVM models well fit the spike statistics of real neural activities. The error bars (SD) were computed across multiple random seeds. **A.** Graphical illustrations of the processing flows of baselines and TeCoS-LVM models. **B.** Heatmaps of the real (Data) and predicted firing rates on Movie 2 Retina 2 test data. The TECOS-LVM Noisy model precisely reproduces the firing rate patterns of real neural data. **C.** Histograms of firing rate Pearson correlation coefficients on test data of four datasets. **D.** Autocorrelograms acquired on test data of four datasets. While the baselines fail, the TeCoS-LVM models accurately capture the spike autocorrelations. **E.** Recorded and predicted activities of two representative neurons responding to repeated trials of a randomly selected test segment of Movie 1 Retina 1 data.

## 4.4 Results

### 4.4.1 TeCoS-LVM Models Accurately Fit Real Spike Activities and Statistics

As shown in Fig. 2B, 2C, TeCoS-LVM models can effectively fit the recorded firing rates. Especially, `TeCoS-LVM Noisy` can reproduce real firing rates more accurately than baselines. Furthermore, we verified the potential information loss caused by using trial-averaged statistics as the optimization target. While the firing rate target may allow models to learn coarse-grained features, it does not necessarily yield optimal capturing of fine-grained features such as spike autocorrelations. As shown by the autocorrelograms in Fig. 2D, the baselines failed to capture the spike autocorrelation

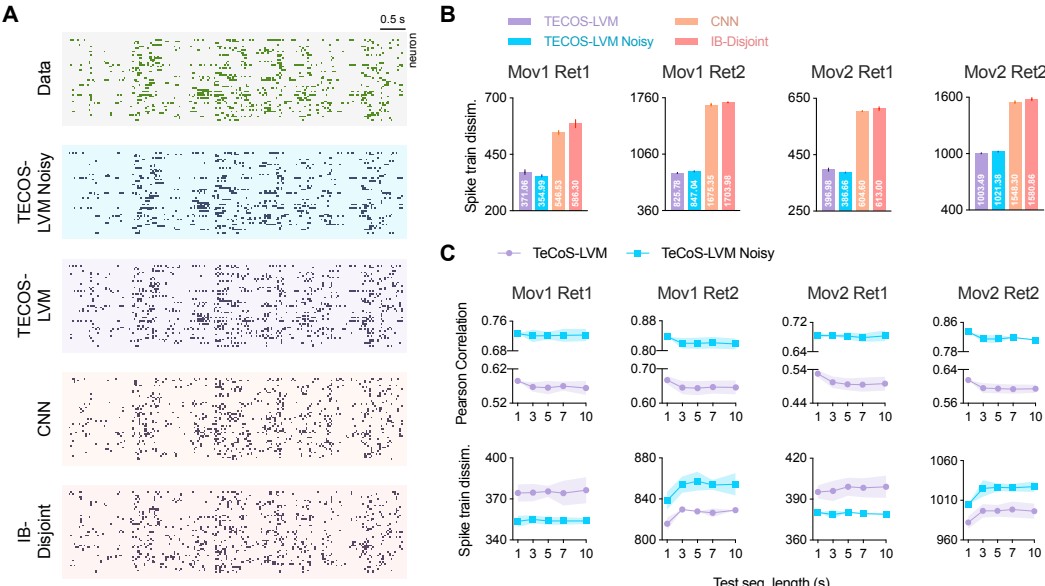

Figure 3: TeCoS-LVM models synthesize realistic neural activities and generalize well to larger time scales. **A.** Examples of the predicted spike trains on a Movie 1 Retina 2 test data clip. The spike trains generated by the TeCoS-LVM models are closer to the recorded spike trains than those of the baselines. **B.** Histograms of spike train dissimilarities on test data. The dissimilarities acquired using TeCoS-LVM models are significantly lower than those of baselines. **C.** TeCoS-LVM models learn general temporal dependencies. We ran TeCoS-LVM models trained using 1-second sequences on longer test sequences and observed that extending the length of the test sequence only leads to minor performance drops.

feature accurately. By contrast, TeCoS-LVM models reproduced the spike autocorrelations more precisely. The TeCoS-LVM models also outperformed other models in synthesizing more realistic spike activities (Fig. 3A&B). Our results suggest that TeCoS-LVM models can accurately fit real spike activities and statistics. In particular, the `TeCoS-LVM Noisy` model significantly outperforms baselines on all metrics.

We also noticed that including neuronal noise is important for modeling neural activities. Because of the absence of neuron-level randomness, the `TeCoS-LVM` model failed to reproduce trial-to-trial variability (Fig. 2E) in real neurons. Consequently, as shown in Fig. 2B, `TeCoS-LVM`'s firing rate predictions (obtained by averaging over multiple runs) are less smooth than others, thus negatively impacting the firing rate correlations. That explains why the `TeCoS-LVM` achieved lower spike train dissimilarities on some data (Movie 1 Retina 2, Movie 2 Retina 2 in Fig. 3B) but still lads behind other methods in terms of the firing rate metric (Fig. 2C).

**Evaluation using more spike distances** The TeCoS-LVM models are specifically optimized to minimize the discrepancy in the PSP kernel space, making them naturally advantageous in comparing PSP-MMD-based spike train dissimilarity. To ensure a more fair assessment of the spike train prediction quality, we have considered several different spike train distances. Specifically, we computed van Rossum, Victor-Purpura, and SPIKE distances (refer to Appendix F for more details) between the predicted spike trains with the recorded ones. We observed consistent and significant improvements in the TeCoS-LVM models over two state-of-the-art baselines, as shown by results in Table 1. We also noticed that when considering these spike train distances, the performance of `TeCoS-LVM Noisy` is slightly affected due to the noise-perturbed neuronal dynamics, which accords with results in Fig. 3B. However, the overall performance of `TeCoS-LVM Noisy` surpasses the baselines by a large margin and, moreover, effectively reproduces the variability in neural processing.

### 4.4.2 Train Short, Test Long: Learned TeCoS-LVM Models Generalize to Longer Time Scales

TeCoS-LVM models completely exclude the time dimension from parameter space and, therefore, can handle input stimuli sequences of any duration without being limited to the length of training data (1 second here). We evaluated the temporal scalability of TeCoS-LVM models by running them on

longer test data sequences. The firing rate correlation coefficients and spike train dissimilarities were calculated to measure the performance changes. TeCoS-LVM models consistently produce accurate coding results at different time scales. Results in Fig. 3C show that increasing the length of the test sequence only brings slight performance drops, as evidenced by the minor decrease in the firing rate correlations and little increase in the spike train dissimilarities. This demonstrates that TeCoS-LVM models learn general (multi-time-scales) temporal dependencies from short-sequence training.

Table 1: Evaluation results using SPIKE, Victor-Purpura, and van Rossum spike train distances, results reported here are averaged across multiple trials.

| DISTANCE TYPE | SPIKE | Victor-Purpura | van Rossum | SPIKE | V.-P. | van Rossum |
|---|---|---|---|---|---|---|
| MODEL \ DATA | MOVIE1 RETINA1 | | | MOVIE1 RETINA2 | | |
| TeCoS-LVM Noisy (This work) | 0.155 | 14.024 | 238.614 | 0.116 | 21.599 | 425.871 |
| TeCoS-LVM (This work) | 0.124 | 12.835 | 127.346 | 0.111 | 18.182 | 150.445 |
| McIntosh NeurIPS-16 | 0.207 | 19.601 | 376.822 | 0.220 | 39.168 | 2672.211 |
| Rahmani NeurIPS-22 | 0.224 | 21.916 | 394.020 | 0.219 | 39.075 | 2276.706 |
| MODEL \ DATA | MOVIE2 RETINA1 | | | MOVIE2 RETINA2 | | |
| TeCoS-LVM Noisy (This work) | 0.162 | 14.412 | 553.510 | 0.153 | 28.441 | 1135.805 |
| TeCoS-LVM (This work) | 0.128 | 12.693 | 308.784 | 0.123 | 22.666 | 574.298 |
| McIntosh NeurIPS-16 | 0.212 | 22.713 | 1650.823 | 0.221 | 39.261 | 2638.964 |
| Rahmani NeurIPS-22 | 0.204 | 22.271 | 1615.934 | 0.221 | 38.378 | 2244.981 |

Figure 4: Visualizations of hidden state and latent space dynamics. **A.** The evolution of $\|\mathbf{h}_t\|$ over time after initialization. **B.** SSIM curve of neighboring stimuli and cosine distance curves of adjacent hidden state vectors of TeCoS-LVM models. **C.** Scatterplots of the Pearson correlation coefficients between hidden state cosine distance and stimuli SSIM. **D.** Model performances under different weighting factor $\beta$s, obtained on Movie 1 Retina 2 test data. **E.** Visualizations of $\mathbf{z}_t$ dynamics in `TeCoS-LVM Noisy` models learned using different levels of $\beta$. **F.** Model performances with and without TeCo (shorthand of temporal conditioning operation). We increased the number of parameters for models without TeCo to be approximately on par with the standard models.

### 4.4.3 Hidden State Analyses: Exploring Memory Mechanisms in TeCoS-LVMs

Interestingly, TeCoS-LVM models' predictions are less accurate for that short period after initialization, likely due to the model's sensory memory requiring time to accumulate. We observed that it takes at most 0.5 seconds for the L2 norm of the hidden state to reach the average value of consecutive runs on a long sequence (Fig. 4A). As a result, we set a warmup period of 0.5 seconds for all our tests. We further investigated the memory mechanism in TeCoS-LVM models, which is implemented by the hidden state update. To this end, we quantified the variation of stimuli by computing the Structural Similarity (SSIM) of adjacent stimuli ($\mathbf{x}_t$ and $\mathbf{x}_{t-1}$). If the change in stimuli at a certain moment is drastic, then the SSIM at this moment is low. For the sensory system, the stimulus it receives at that time is quite novel or surprising [61]. We also measured the memory update magnitude by calculating the cosine distance of adjacent hidden state vectors ($\mathbf{h}_t$ and $\mathbf{h}_{t-1}$). As shown in Fig. 4B,

C, the two have a strong negative correlation. This suggests that the TeCoS-LVM models perform more significant memory updates (indicated by a large cosine distance) when facing highly dynamic stimuli (low SSIM); only minor updates are performed when stimuli are relatively stable. This is consistent with previous sensory neuroscience findings suggesting that the sensory circuits focus more on unpredictable or surprising events. When repeatedly exposed to an initially novel stimulus, the neural processing becomes less active [62, 63, 61, 64].

#### 4.4.4 Latent Space Dynamic Analyses

We next explored the latent variables under different weighting factor $\beta$ settings. Setting a proper $\beta$ value can lead to richer latent variable dynamics. It is more advantageous to show how they relate to factors like anatomy and behavior through correlation analysis, thereby interpreting the inferred latent variables. The compressive loss term forces the latent variable to act like a minimal sufficient statistic of stimuli for predicting neural responses. Therefore, tiny $\beta$ values correspond to weak regularization provided by the compressive term. In this case, the latent representation learns to be more deterministic (Fig. 4E-Left). This prevents the model from benefiting from the regularization brought by the compressive loss term, resulting in poorer performance compared to moderate $\beta$ values (Fig. 4D). When using a large $\beta$, we observed that the latent variables seem to have lost some temporal information in the stimuli (Fig. 4E-Right). And, if $\beta$ is further increased, the model cannot obtain enough stimuli information to predict the responses, resulting in a sharp decline in performance (Fig. 4D).

#### 4.4.5 Functional Implication Analyses by In-silico Ablating

**Temporal conditioning operation** We then evaluated the performance of TeCoS-LVM models without temporal conditioning operations, which is a reduced variant with an isotropic Gaussian prior (see also Appendix Fig. 5). Results in Fig. 4F show that the TeCo operation significantly improves model performance by exploiting the temporal dependencies. This suggests that visual coding requires the integration and manipulation of temporally dispersed information from continuous streams of stimuli.

**Spiking hidden neurons** We conducted experiments to investigate the effect of employing spiking neurons in modeling neural activity to natural stimuli (see also Appendix H). Specifically, we replaced all spiking hidden neurons in the TeCoS-LVM models with LIF-Rate neurons[65] while retaining only the output spiking neurons so that the training pipeline remains unchanged. The LIF-Rate neurons maintain the neuronal recurrency by using the same membrane potential update as LIF and Noisy LIF neurons. This allows us to directly assess the effect of using spiking neurons by ablating them. Our experimental results show that the use of spiking neurons can significantly enhance the performance of computational models. As can be seen from Table 2, all performance indicators have significantly decreased after replacing the spiking hidden units with rate ones. Previous studies have pointed out that natural stimuli have a sparse latent structure [66]; therefore, using spiking hidden units may better fit the latent structure of natural stimuli by utilizing sparse spike representation. Also, previous research [67] indicated that the coding method of SNNs can lead to highly competitive results, which is consistent with the conclusion of this part of our experiment.

Table 2: Ablation results of using spiking hidden neurons. An ↑ indicates that the higher the value, the better, while a ↓ suggests the opposite. Results reported are averaged across multiple trials.

| | | Spiking hidden units | CC (↑) | Spike Train Dissim. (↓) | SPIKE (↓) | Victor-Purpura (↓) | van Rossum (↓) |
|---|---|---|---|---|---|---|---|
| Mov1 Ret1 | TeCoS-LVM | Yes | **0.579** | **371.057** | **0.124** | **12.835** | **127.346** |
| | | No | 0.254 | 850.418 | 0.259 | 45.117 | 3416.557 |
| | TeCoS-LVM Noisy | Yes | **0.728** | **354.989** | **0.155** | **14.024** | **238.614** |
| | | No | 0.653 | 370.099 | 0.167 | 14.805 | 291.706 |
| Mov2 Ret2 | TeCoS-LVM | Yes | **0.616** | **1003.489** | **0.123** | **22.666** | **574.298** |
| | | No | 0.471 | 1273.267 | 0.180 | 35.080 | 1890.910 |
| | TeCoS-LVM Noisy | Yes | **0.822** | **1021.384** | **0.153** | **28.441** | **1135.805** |
| | | No | 0.748 | 1078.830 | 0.159 | 29.249 | 1144.087 |

## 5 Related works

This work builds on previous research on modeling neural responses to visual scenes. Early attempts included Linear-nonlinear (LN) [68, 69] and Generalized Linear Models (GLMs) [2, 7]. However,

these models have limited capabilities and fail when faced with more complex stimuli [8, 9]. A promising way is to leverage powerful neural networks. One attractive advantage of this approach is that it may eliminate the need to specify spike statistics explicitly. McIntosh et al. [9] proposed a convolutional neural network (CNN) approach that outperformed LN and GLM baselines by a large margin. Some researchers have also investigated CNN variants with a recurrent layer [9, 15]. Batty et al. [12] proposed a hybrid model that combines GLMs and recurrent neural networks (RNNs) to separate spatial and temporal processing components in neural coding. Our model also has specified structures that deal with the time dimension, but the temporal and spatial processing are tightly integrated. Generative adversarial networks (GANs) have also been used to synthesize realistic neural activities [14], but this method cannot be used to predict neural responses given the stimuli. Mahuas et al. [70] introduced a novel two-step strategy to improve the basic GLMs. Bellec et al. [71] introduced a model with spiking response model (SRM) neurons and proposed a sample and measure criteria in addition to the traditional Poisson likelihood objective. More recently, Rahmani et al. [21] introduced the Gaussian Process prior and variational information bottleneck to LVMs for modeling retinal responses. Compared to biological coding systems, these methods have two aspects of authenticity missing (Fig. 2A). Firstly, most of these methods target trial-averaged firing rates. Secondly, they cannot directly process long stimuli sequences in a *natural paradigm* but must divide them into segments of pre-defined lengths to be processed separately.

**Latent variable models**    LVMs are popular tools for investigating and modeling neural response [18, 19, 72]. Some studies suggest that low-dimensional latent factors can effectively represent high-dimensional neural activities [16, 17]. In recent works, LVMs demonstrated promising results in uncovering the low-dimensional structure underlying complex neural activities in the motor region [20, 23, 22]. Also, LVMs have shown promise in visual neural coding modeling [18, 19, 21].

**Conditioning**    Conditioning is an effective technique to integrate information from multiple sources and realize adaptive information gating. Typically, conditioning methods rely on external information sources like labels [73] or outputs from other models [74, 75]. Some studies have demonstrated that direct conditioning on previous states can also significantly improve performance, especially in a sequential processing regime [49, 76]; these approaches are often termed self-conditioning or temporal conditioning. Notably, the temporal conditioning design exposed here is loosely inspired by the Tolman-Eichenbaum Machine [49], where the conditional operations ensure that the model profits from a sequential learning paradigm.

## 6    Discussion

This work presents the TeCoS-LVM (temporal conditioning spiking LVM) models. Our approach is formalized within the information bottleneck framework of latent variable models, inspired by the efficient coding theory [34, 38]. TeCoS-LVM models can directly produce neural response sequences in real-time rather than repeatedly producing single-step predictions, thus faithfully simulating the biological coding process. Using the retinal response to natural scenes as an example, we showed that TeCoS-LVM models effectively fit spike statistics, spike features, and recorded spike trains. In particular, TeCoS-LVM models that incorporate Noisy LIF neurons significantly exceed high-performance baselines. Also, learned TeCoS-LVM models generalize well to longer time scales, indicating that they can learn general temporal features from short-sequence training. Overall, TeCoS-LVM models effectively capture key features of biological coding systems while remaining computationally tractable.

**Outlook**    Although we use visual coding as an example to demonstrate the impressive performance of TeCoS-LVM models in this article, the models exposed here *can be easily applied to sensory data of other modalities*. As such, TeCoS-LVM demonstrates a promising framework for building computational accounts for various sensory neural circuits and will enable richer models of complex neural computations in the brain.

**Limitations**    As an accurate model that relates stimulus-driven responses, from a *neuroconnectionsim* perspective [77], the TeCoS-LVM model aims to provide neuroscientific insights and understandings at the computational level. Compared to previous endeavors, TeCoS-LVM takes spike-based neural computation and natural processing paradigms into account. Nonetheless, while the TeCoS-LVM model makes predictions at the level of retinal cells, its fundamental principles are at a computational level, so it is only partially biophysically realistic.

## Acknowledgement

This work is supported by the National Key Research and Development Program of China under Grant 2020AAA0105900 and the National Natural Science Foundation of China under Grant 62236007, Grant U2030204.

The authors are grateful for the generous support from Professor Arno Onken from the University of Edinburgh. The authors would also like to acknowledge anonymous reviewers and chairs of NeurIPS 2023 for providing insightful comments to help improve this work.

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

**Appendix of** *Temporal Conditioning Spiking Latent Variable Models of the Neural Response to Natural Visual Scenes*

## A    Hidden State and Latent Space Experiments

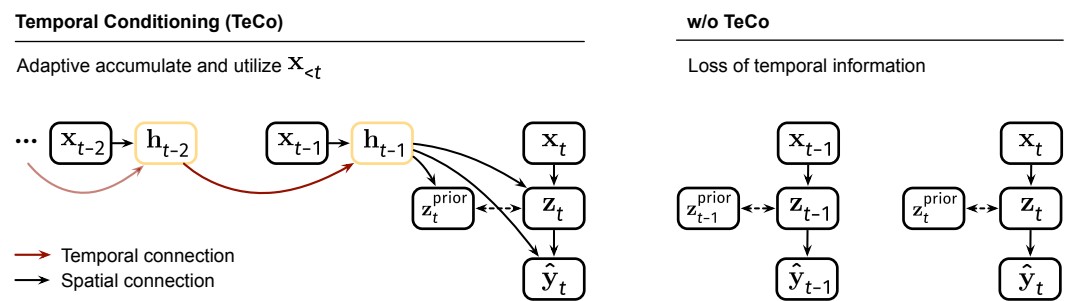

Figure 5: Graphical illustration of the temporal conditioning operation (TeCo). After completely excluding the temporal dimension from the model parameter space, we introduced the temporal conditioning operation to handle the temporal information. In particular, this operation enables *memory-dependent processing* as in biological coding circuits.

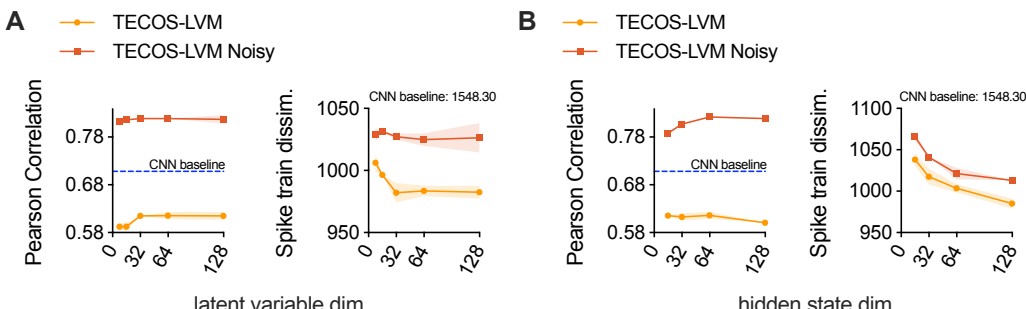

Figure 6: Performances under different hidden state and latent space dimension settings on Movie 2 Retina 2 data. For hidden state experiments, the latent space dimension is set to 32. And for latent space experiments, the hidden state dimension is 64. The error bars (SD) were calculated via various random seeds.

To further study the sensitivity to the choice of hyperparameters, we evaluated the performance of TeCoS-LVM models under different hidden state and latent space dimensionality settings. The results, as shown in Appendix Fig. 6A, indicate that increasing the latent space dimension improved performance. Still, further increases (larger than 32) had little effect when the latent space dimension was large. This also suggests that the number of latent factors for the visual stimuli coding task we considered is not very large. On the other hand, increasing the hidden state dimension enhances sensory memory capacity and benefits temporal conditioning operations. As shown in Appendix Fig. 6B, the performance does not increase significantly when hidden state dimensionality increases to a certain extent (around 64). In particular, although the spike train dissimilarity decreases, the firing rate correlation score almost no longer increases. This is consistent with our observation in the main text, namely, a lower spike train dissimilarity does not always indicate a higher firing rate correlation score (refer to Fig. 2,3). The results in Appendix Fig. 6B also indicate that the proposed temporal conditioning mechanism can effectively utilize sensory memory and achieve good results even when the hidden state dimension is limited.

## B    Surrogate Gradient Learning in Spiking Networks

In conventional SNN Surrogate Gradient Learning (SGL, pseudo derivative, derivative approximation) [29], the derivative of the firing function $\partial o/\partial u$ is replaced by a smooth function (pseudo derivative

function) SG to mesh with the backpropagation scheme [78]. This ad-hoc technique in SNN research is popular, particularly for large-scale networks. It allows for compatibility with popular automatic differentiation packages such as PyTorch and TensorFlow, simplifying the implementation of SNNs. This surrogate gradient function can be a triangular, rectangular (gate), sigmoidal, or ERF function [78]. In SGL, the gradient $g_l$ w.r.t. synaptic weights of layer $l$ is calculated by

$$\text{SGL: } \hat{g}_l = \sum_m \nabla_{\theta_l} u_t^{l,m} \underbrace{\text{SG}(u_t^{l,m} - v_{\text{th}})}_{\text{Surrogate the exact derivative } \partial o_t^{l,m}/\partial u_t^{l,m}} \nabla_{o_t^{l,m}} \mathcal{L}_t, \tag{10}$$

where $l, m$ denotes neuron $m$ in layer $l$, and $\mathcal{L}_t$ is the instant loss value.

## C  Leveraging Noisy Spiking Neural Models

Here, we use the implementation in [53] to leverage the power of noisy spiking neural models. Spiking neurons with noisy neuronal dynamics have been extensively studied in prior literature [25]. Recent research of Ma et al. [53] extended them to larger networks by providing a general formularization and demonstrating their computational advantages theoretically and empirically. The Noisy LIF presented here is based on previous works that use diffusive approximation [79, 80, 25], where the sub-threshold dynamic is described by the Ornstein-Uhlenbeck process:

$$\tau_m \frac{du}{dt} = -(u - u_{\text{reset}}) + RI(t) + \xi(t), \text{ eq. } du = -(u - u_{\text{reset}})\frac{dt}{\tau_m} + RI(t)\frac{dt}{\tau_m} + \sigma dW_t, \tag{11}$$

the white noise $\xi$ is a stochastic process, $\sigma$ is the amplitude of the noise and $dW_t$ are the increments of the Wiener process in $dt$ [25]. As $\sigma dW_t$ are random variables drawn from a zero-mean Gaussian, this formulation is directly applicable to discrete-time simulations. Specifically, using the Euler-Maruyama method, we get a Gaussian noise term added on the right-hand side of the noise-free LIF dynamic. Without loss of generality, we extend the additive noise term in the discrete form to general continuous noise [81], the sub-threshold dynamic of Noisy LIF can be represented as:

$$\text{Noisy LIF sub-threshold dynamic: } u_t = \tau u_{t-1} + I_t + \epsilon, \tag{12}$$

where $I_t$ is the input, the noise $\epsilon$ is independently drawn from a known distribution and satisfies $\mathbb{E}[\epsilon] = 0$ and $p(\epsilon) = p(-\epsilon)$. The constant $\tau$ here combines the simulation timestep length and the real membrane decay $\tau_m$, which is a simplification when the timestep we cope with is fixed. This work considers the Gaussian noise $\epsilon \sim \mathcal{N}(0, 0.2^2)$.

The membrane potentials and spike outputs become random variables due to random noise injection. Leveraging noise as a medium, we naturally obtain the firing probability distribution of Noisy LIF based on the threshold firing mechanism [53]:

$$\mathbb{P}[\text{firing at time } t] = \mathbb{P}\underbrace{[u_t + \epsilon > v_{\text{th}}]}_{\text{Threshold-based firing}} = \underbrace{\mathbb{P}[\epsilon < u_t - v_{\text{th}}] \triangleq F_\epsilon(u_t - v_{\text{th}})}_{\text{Cumulative Distribution Function definition}},$$

where $F$ denotes the cumulative distribution function. Therefore, we have that,

$$o_t = \begin{cases} 1, \text{with probability} & F_\epsilon(u_t - v_{\text{th}}), \\ 0, \text{with probability} & (1 - F_\epsilon(u_t - v_{\text{th}})). \end{cases} \tag{13}$$

The expressions above exemplify how noise acts as a resource for computation [82]. Thereby, we can formulate the firing process of Noisy LIF as [53]

$$\text{Noisy LIF probabilistic firing: } o_t \sim \text{Bernoulli}(F_\epsilon(u_t - v_{\text{th}})), \tag{14}$$

Specifically, it relates to previous literature on noise escape models, in which the difference $u - v_{\text{th}}$ governs the neuron firing probabilities [83, 79, 25]. In addition, Noisy LIF employs the same resetting mechanism as the LIF model.

### C.1  Noise-Driven Learning in Networks of Noisy LIF Neurons

The Noise-Driven Learning (NDL) rule [53] in networks of Noisy LIF neurons is a theoretically sound general form of Surrogate Gradient Learning. In particular, the gradient w.r.t to synaptic

weights in layer $l$ is computed by

$$\text{NDL: } \hat{g}_l = \sum_m \underbrace{\nabla_{\theta_l} u_t^{l,m}}_{\text{Pre-synaptic factor}} \overbrace{F'_\epsilon(u_t^{l,m} - v_{\text{th}})}^{\text{Post-synaptic factor}} \underbrace{\nabla_{o_t^{l,m}} \mathcal{L}_t}_{\text{Global learning signal}} , \tag{15}$$

where the superscript $l, m$ denotes neuron $m$ in layer $l$, $\mathcal{L}$ is the loss value. Here, the post-synaptic factor in NDL is calculated by the probability distribution function of the postsynaptic neuron's membrane potential noise.

As shown in Equation 15, NDL is well-compatible with the backpropagation computation paradigm in trending libraries like PyTorch. Therefore, we can wrap the inference and learning of Noisy LIF neurons into a module. By replacing the original LIF neuron module (with Surrogate Gradient Learning) with the Noisy LIF module (with NDL), we can easily implement noisy spiking neural networks of arbitrary architectures in a *plug-and-play* manner. An example can be found at `https://github.com/genema/Noisy-Spiking-Neuron-Nets`.

# D    Derivation of the Optimization Objective of TeCoS-LVM Models

**Basic formulation – from an efficient coding [38] perspective**    We denote a sequence of visual stimuli as $\mathbf{x} = (\mathbf{x}_t)_{t=1\cdots T}$, where $\mathbf{x}_t \in \mathbb{R}^{\dim[\mathbf{x}_t]}$, $\dim[\mathbf{x}_t]$ stands for the dimension of $\mathbf{x}_t$. Similarly, we denote neural population response (target) as $\mathbf{y} = (\mathbf{y}_t) \in \{0,1\}^{T \times \dim[\mathbf{y}_t]}$, where $\dim[\mathbf{y}_t]$ denotes the number of retinal ganglion cells (RGCs). At each timestep $t$, one high-dimensional visual stimulus $\mathbf{x}_t$ is received, and we want to predict the neural population response $\mathbf{y}_t$. This is implemented by an LVM which first compresses the visual stimuli into a low-dimensional latent representation $\mathbf{z}_t \in \mathbb{R}^{\dim[\mathbf{z}_t]}$, and then decodes the neural population response from it. Inspired by the information compression feature in the neural coding process [34, 35, 36, 37], we further encourage this LVM to construct a latent space in which $\mathbf{z}$ have maximal predictive power regarding $\mathbf{y}$ while being maximally compressive about $\mathbf{x}$. Therefore, our target to model the neural coding of visual stimuli turns into an optimization problem within the IB framework. *Note that stimulus from other modalities can be processed in a similar vein, but here we use the visual case as an example.*

Following previous IB literature [31, 32, 33], we assume a factorization of the joint distribution as follows,

$$p(\mathbf{x}, \mathbf{y}, \mathbf{z}) = p(\mathbf{z}|\mathbf{x}, \mathbf{y})p(\mathbf{y}|\mathbf{x})p(\mathbf{x}) = p(\mathbf{z}|\mathbf{x})p(\mathbf{y}|\mathbf{x})p(\mathbf{x}), \tag{16}$$

namely, we assume a Markov chain $\mathbf{y} \leftrightarrow \mathbf{x} \leftrightarrow \mathbf{z}$, which implies $p(\mathbf{z}|\mathbf{x}, \mathbf{y}) = p(\mathbf{z}|\mathbf{x})$, indicating that the latent representation $\mathbf{z}$ cannot directly depend on the target response $\mathbf{y}$. Recall that, according to the IB principle [31], our objective has the form

$$\text{IB objective: } \max[\underbrace{I(\mathbf{z}, \mathbf{y})}_{\text{Predictive term}} \underbrace{-\beta I(\mathbf{z}, \mathbf{x})}_{\text{Compressive term}}]. \tag{17}$$

The predictive term encourages predictive power, while the compressive term enforces information compression. And it is equivalent to minimizing a loss function $-I(\mathbf{z}, \mathbf{y}) + \beta I(\mathbf{z}, \mathbf{x})$.

Let us examine the predictive term $I(\mathbf{z}, \mathbf{y})$ first. The mutual information between $\mathbf{z}$ and $\mathbf{y}$ is given by

$$\begin{aligned} I(\mathbf{z}, \mathbf{y}) &= \int \mathrm{d}\mathbf{y}\mathrm{d}\mathbf{z}\, p(\mathbf{y}, \mathbf{z}) \log \frac{p(\mathbf{y}, \mathbf{z})}{p(\mathbf{y})p(\mathbf{z})} \\ &= \int \mathrm{d}\mathbf{y}\mathrm{d}\mathbf{z}\, p(\mathbf{y}, \mathbf{z}) \log \frac{p(\mathbf{y}|\mathbf{z})}{p(\mathbf{y})}. \end{aligned} \tag{18}$$

According to the assumed Markov chain (16), the likelihood $p(\mathbf{y}|\mathbf{z})$ is defined by

$$p(\mathbf{y}|\mathbf{z}) = \int \mathrm{d}\mathbf{x}\, p(\mathbf{x}, \mathbf{y}|\mathbf{z}) = \int \mathrm{d}\mathbf{x}\, p(\mathbf{y}|\mathbf{x})p(\mathbf{x}|\mathbf{z}) = \int \mathrm{d}\mathbf{x}\, p(\mathbf{y}|\mathbf{x}) \frac{p(\mathbf{z}|\mathbf{x})p(\mathbf{x})}{p(\mathbf{z})}, \tag{19}$$

and is approximated by a variational decoder $p(\mathbf{y}|\mathbf{z}; \psi^{\text{dec}})$ in our case. Given the fact that $\text{KL}[p(\mathbf{y}|\mathbf{z})\|p(\mathbf{y}|\mathbf{z}; \psi^{\text{dec}})] \geq 0$, we can write

$$\begin{aligned} &\int \mathrm{d}\mathbf{y}\, p(\mathbf{y}|\mathbf{z}) \log \frac{p(\mathbf{y}|\mathbf{z})}{p(\mathbf{y}|\mathbf{z}; \psi^{\text{dec}})} \geq 0 \\ \Rightarrow &\int \mathrm{d}\mathbf{y}\, p(\mathbf{y}|\mathbf{z}) \log p(\mathbf{y}|\mathbf{z}) \geq \int \mathrm{d}\mathbf{y}\, p(\mathbf{y}|\mathbf{z}) \log p(\mathbf{y}|\mathbf{z}; \psi^{\text{dec}}). \end{aligned} \tag{20}$$

Therefore,

$$I(\mathbf{z}, \mathbf{y}) \geq \int \mathrm{d}\mathbf{y}\mathrm{d}\mathbf{z}\, p(\mathbf{y}, \mathbf{z}) \log \frac{p(\mathbf{y}|\mathbf{z}; \psi^{\mathrm{dec}})}{p(\mathbf{y})}$$

$$= \int \mathrm{d}\mathbf{y}\mathrm{d}\mathbf{z}\, p(\mathbf{y}, \mathbf{z}) \log p(\mathbf{y}|\mathbf{z}; \psi^{\mathrm{dec}}) + H(\mathbf{y}). \tag{21}$$

Since the target information entropy $H(\mathbf{y})$ is independent of the optimization procedure of the parametric model, it can be ignored. Thus, $\max I(\mathbf{z}, \mathbf{y}) = \max \int \mathrm{d}\mathbf{y}\mathrm{d}\mathbf{z}\, p(\mathbf{y}, \mathbf{z}) \log p(\mathbf{y}|\mathbf{z}; \psi^{\mathrm{dec}})$. By Eq. 16, $p(\mathbf{y}, \mathbf{z}) = \int \mathrm{d}\mathbf{x}\, p(\mathbf{x}) p(\mathbf{y}|\mathbf{x}) p(\mathbf{z}|\mathbf{x})$, therefore,

$$\max I(\mathbf{z}, \mathbf{y}) = \max \int \mathrm{d}\mathbf{x}\mathrm{d}\mathbf{y}\mathrm{d}\mathbf{z}\, p(\mathbf{x}) p(\mathbf{y}|\mathbf{x}) p(\mathbf{z}|\mathbf{x}) \log p(\mathbf{y}|\mathbf{z}; \psi^{\mathrm{dec}}). \tag{22}$$

We now consider the compressive term $\beta I(\mathbf{z}, \mathbf{x})$ in the IB objective (17), and we temporally discard the constant factor $\beta$. The mutual information between input stimuli and latent representation is given by

$$I(\mathbf{z}, \mathbf{x}) = \int \mathrm{d}\mathbf{x}\mathrm{d}\mathbf{z}\, p(\mathbf{x}, \mathbf{z}) \log \frac{p(\mathbf{z}|\mathbf{x})}{p(\mathbf{z})}$$

$$= \int \mathrm{d}\mathbf{x}\mathrm{d}\mathbf{z}\, p(\mathbf{x}, \mathbf{z}) \log p(\mathbf{z}|\mathbf{x}) - \int \mathrm{d}\mathbf{z}\, p(\mathbf{z}) \log p(\mathbf{z}). \tag{23}$$

Let $p(\mathbf{z}; \phi^{\mathrm{prior}})$ be a variational approximation to the marginal $p(\mathbf{z})$, because $\mathrm{KL}[p(\mathbf{z})\|p(\mathbf{z}; \phi^{\mathrm{prior}})] \geq 0$, we have that

$$\int \mathrm{d}\mathbf{z}\, p(\mathbf{z}) \log p(\mathbf{z}) \geq \int \mathrm{d}\mathbf{z}\, p(\mathbf{z}) \log p(\mathbf{z}; \phi^{\mathrm{prior}}). \tag{24}$$

With Eq. 23 in tow and using a parametric encoder $q(\mathbf{z}|\mathbf{x}; \psi^{\mathrm{enc}})$, we have the following upper bound:

$$I(\mathbf{z}, \mathbf{x}) \leq \int \mathrm{d}\mathbf{x}\mathrm{d}\mathbf{z}\, p(\mathbf{x}) q(\mathbf{z}|\mathbf{x}; \psi^{\mathrm{enc}}) \log \frac{q(\mathbf{z}|\mathbf{x}; \psi^{\mathrm{enc}})}{p(\mathbf{z}; \phi^{\mathrm{prior}})}. \tag{25}$$

By Eqs. 22, 25, we have a lower bound for the IB objective as follows,

$$I(\mathbf{z}, \mathbf{y}) - \beta I(\mathbf{z}, \mathbf{x}) \geq \int \mathrm{d}\mathbf{x}\mathrm{d}\mathbf{y}\mathrm{d}\mathbf{z}\, p(\mathbf{x}) p(\mathbf{y}|\mathbf{x}) q(\mathbf{z}|\mathbf{x}; \psi^{\mathrm{enc}}) \log p(\mathbf{y}|\mathbf{z}; \psi^{\mathrm{dec}})$$

$$- \beta \int \mathrm{d}\mathbf{x}\mathrm{d}\mathbf{z}\, p(\mathbf{x}) q(\mathbf{z}|\mathbf{x}; \psi^{\mathrm{enc}}) \log \frac{q(\mathbf{z}|\mathbf{x}; \psi^{\mathrm{enc}})}{p(\mathbf{z}; \phi^{\mathrm{prior}})}. \tag{26}$$

Using $\boldsymbol{\theta}$ to denote all the parameters ($\phi^{\mathrm{prior}}, \psi^{\mathrm{enc}}, \psi^{\mathrm{dec}}$ and other learnable parameters, like those of the feature extractor) of the model following the main text, we have that

$$\max_{\boldsymbol{\theta}}[I(\mathbf{z}, \mathbf{y}; \boldsymbol{\theta}) - \beta I(\mathbf{z}, \mathbf{x}; \boldsymbol{\theta})] = \min_{\boldsymbol{\theta}} \mathcal{L}, \tag{27}$$

where

$$\mathcal{L} = \underbrace{- \int \mathrm{d}\mathbf{x}\mathrm{d}\mathbf{y}\mathrm{d}\mathbf{z}\, p(\mathbf{x}) p(\mathbf{y}|\mathbf{x}) q(\mathbf{z}|\mathbf{x}; \psi^{\mathrm{enc}}) \log p(\mathbf{y}|\mathbf{z}; \psi^{\mathrm{dec}})}_{\mathcal{L}^{\mathrm{pred}}:\ \text{encouraging predictive power}}$$

$$+ \beta \underbrace{\int \mathrm{d}\mathbf{x}\mathrm{d}\mathbf{z}\, p(\mathbf{x}) q(\mathbf{z}|\mathbf{x}; \psi^{\mathrm{enc}}) \log \frac{q(\mathbf{z}|\mathbf{x}; \psi^{\mathrm{enc}})}{p(\mathbf{z}; \phi^{\mathrm{prior}})}}_{\mathcal{L}^{\mathrm{comp}}:\ \text{encouraging compression}}. \tag{28}$$

As for now, we have the formulation presented in Eq. 3 in the main text. We proceed to derive the exact loss function for TeCoS-LVM model learning. Following previous literature [32], we can approximate the data distribution $p(\mathbf{x}, \mathbf{y})$ using the empirical data distribution $\frac{1}{T} \sum_{t=1}^{T} \delta(\mathbf{x} - \mathbf{x}_{1:t}) \delta(\mathbf{y} - \mathbf{y}_t)$, where $\delta$ is the dirac delta function. Hence, we have that

$$\mathcal{L} \approx \frac{1}{T} \sum_{t=1}^{T} \Big[ \underbrace{\mathbb{E}_{q(\mathbf{z}_t|\mathbf{x}_{1:t}; \psi^{\mathrm{enc}})}[-\log p(\mathbf{y}_t|\mathbf{z}_t; \psi^{\mathrm{dec}})]}_{\mathcal{L}_t^{\mathrm{pred}}} + \beta \underbrace{\mathrm{KL}[q(\mathbf{z}_t|\mathbf{x}_{1:t}; \psi^{\mathrm{enc}})\|p(\mathbf{z}_t; \phi^{\mathrm{prior}})]}_{\mathcal{L}_t^{\mathrm{comp}}} \Big]$$

$$= \underbrace{\frac{1}{T} \sum_t \mathcal{L}_t^{\mathrm{pred}}}_{\text{Predictive term (total): } \mathcal{L}^{\mathrm{pred}}} + \beta \underbrace{\frac{1}{T} \sum_t \mathcal{L}_t^{\mathrm{comp}}}_{\text{Compressive term (total): } \mathcal{L}^{\mathrm{comp}}}. \tag{29}$$

As we adopt Gaussian distributions in our temporal conditioning prior and encoder, we can analytically compute the compressive loss term $\mathcal{L}^{\mathrm{comp}}$ composed of Kullback-Leibler divergences.

We then turn to the predictive term in our objective (17). As our model directly produces simulated spike trains, we compute the spike train dissimilarity between the prediction $\hat{\mathbf{y}}_{1:t}$ and the real record $\mathbf{y}_{1:t}$ to assess the predictive power of our model directly. This dissimilarity is used as the predictive loss term at each timestep. In particular, we employ the Maximum Mean Discrepancy (MMD) to measure the distance between spike trains. This approach has been proven suitable for spike trains in previous literature [26, 50], following ref. [50], we use a postsynaptic potential (PSP) function kernel for MMD. We employ the first-order synaptic model as the PSP function to capture the temporal dependencies in spike train data effectively [51]. The PSP kernel we shall use is given by

$$\kappa_{\mathrm{PSP}}(\hat{\mathbf{y}}_{1:t}, \mathbf{y}_{1:t}) = \sum_{\tau=1}^{t} \mathrm{PSP}(\hat{\mathbf{y}}_{1:\tau})\mathrm{PSP}(\mathbf{y}_{1:\tau}), \text{ where } \mathrm{PSP}(\mathbf{y}_{1:\tau}) = (1 - \frac{1}{\tau_s})\mathrm{PSP}(\mathbf{y}_{1:\tau-1}) + \frac{1}{\tau_s}\mathbf{y}_\tau, \quad (30)$$

here $\tau_s$ is a synaptic time constant set to 2 by default. We can write the (squared) PSP kernel MMD between the empirical data distribution and the predictive distribution as

$$\mathrm{MMD}[p_{\psi^{\mathrm{dec}}}(\hat{\mathbf{y}}_{1:t}), p(\mathbf{y}_{1:t})]^2 = \sum_{\tau=1}^{t} \left\| \mathbb{E}_{\hat{\mathbf{y}}_{1:\tau} \sim p_{\psi^{\mathrm{dec}}}}[\mathrm{PSP}(\hat{\mathbf{y}}_{1:\tau})] - \mathbb{E}_{\mathbf{y}_{1:\tau} \sim p}[\mathrm{PSP}(\mathbf{y}_{1:\tau})] \right\|^2 \quad (31)$$

In practice, we approximate the spike train dissimilarity, which is measured by the squared PSP kernel MMD in Eq. 31 by $\sum_\tau \|\mathrm{PSP}(\hat{\mathbf{y}}_{1:\tau}) - \mathrm{PSP}(\mathbf{y}_{1:\tau})\|^2$ [50]. Therefore, the predictive loss term is given by

$$\mathcal{L}_t^{\mathrm{pred}} = \sum_{\tau=1}^{t} \|\mathrm{PSP}(\hat{\mathbf{y}}_{1:\tau}) - \mathrm{PSP}(\mathbf{y}_{1:\tau})\|^2 . \quad (32)$$

Together with the compressive term $\mathcal{L}_t^{\mathrm{comp}} = \mathrm{KL}[q_{\psi^{\mathrm{enc}}}(\mathbf{z}_t)\|p_{\phi^{\mathrm{prior}}}(\mathbf{z}_t)]$, by Eq. 29, we can calculate the loss function and optimize TeCoS-LVM models. To allow direct backpropagation through a single sample of the stochastic latent representation, we use the reparameterization trick as described in [84].

# E   Data Description

| Movie 1 ("salamander movie") | Movie 2 ("wildlife movie") |
|---|---|
| 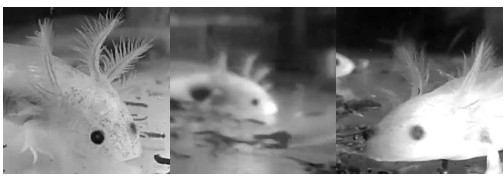 | 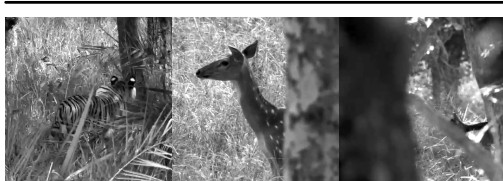 |

Figure 7: Example frames from Movie 1 and Movie 2 in the data we used.

We perform evaluations and analyses on real neural recordings from RGCs of dark-adapted axolotl salamander retinas. The original dataset [57] contains the spike neural responses (collected using multi-electrode arrays [85, 86]) of two retinas on two movies. Movie 1 contains natural scenes of salamanders swimming in the water. Movie 2 contains complex natural scenes of a tiger on a prey hunt. Both movies were roughly 60 $s$ long and were discretized into bins of 33 $ms$. All movie frames were converted to grayscale with a resolution of 360 pixel×360 pixel at 7.5 $\mu m$×7.5 $\mu m$ per pixel, covering a 2700 $\mu m$×2700 $\mu m$ area on the retina. For retina 1, we have 75 repetitions for movie 1 and 107 repetitions for movie 2. For retina 2, we have 30 and 42 repetitions for movie 1 and movie 2, respectively. Some example frames are shown in Appendix Fig. 7.

# F Metrics, Features used in Evaluations and Visualizations

## F.1 Evaluation Metrics

**Pearson correlation coefficient (Pearson CC, CC)**   This metric evaluates the model performance by calculating the Pearson correlation coefficient between the recorded and predicted firing rates [9, 14, 15]. The higher the value, the better the performance. For spike-output TeCoS-LVM models, the firing rates are calculated using 20 repeated trials.

**Spike train dissimilarity (Spike train dissim.)**   This metric assesses the model performance by computing the dissimilarity between recorded and predicted spike trains. A lower value indicates better model performance. We use the MMD with a first-order PSP kernel [51, 52] to measure the spike train dissimilarity [27, 50, 58]. The first-order PSP function is given by $\text{PSP}(\mathbf{y}_{1:t}) = (1 - \frac{1}{\tau_s})\text{PSP}(\mathbf{y}_{1:t-1}) + \frac{1}{\tau_s}\mathbf{y}_t$, where $\tau_s$ is a synaptic constant and is set to 2. Given a recorded spike train $\mathbf{y}_{1:T}$ and a predicted spike train $\hat{\mathbf{y}}_{1:T}$, this metric is calculated by $\sum_{t=1}^{T} \|\text{PSP}(\mathbf{y}_{1:t}) - \text{PSP}(\hat{\mathbf{y}}_{1:t})\|^2$. Because of the variability of neural activities, we randomly selected ten (trials) recorded spike trains and used their average value in our evaluations.

**van Rossum distance (van Rossum)**   This spike train distance was introduced in ref. [87], where the discrete spike trains are convolved by an exponential kernel $\texttt{Heaviside}(t)\exp(-t/\tau_R)$, here we use $\tau_R = 10$. The final scores are computed by averaging results calculated using ten recorded spike trains.

**Victor-Purpura distance (V.-P.)**   This spike train distance [88] measures the dissimilarity between two spike trains by summing up the minimum cost of transforming one spike train into the other by insertion, deletion, and shifting operations. We use the average results from ten trials as the final metric.

**SPIKE distance (SPIKE)**   The SPIKE distance [89] is a time-scale independent metric for quantifying the dissimilarity between spike trains. Its value is bounded in the interval $[0, 1]$, and zero is obtained only for perfectly identical trains.

## F.2 Spike Feature

**Spike autocorrelogram**   The spike autocorrelogram is computed by counting the number of spikes that occur around each spike within a predefined time window [14, 9]. The resulting trace is then normalized to its maximum value (which occurs at the origin of the time axis by construction). In the main text, the maximum value is set to zero for better visualization and comparison.

# G Experimental Details

## G.1 Experimental Platform

The models are implemented using Python and PyTorch. Our experiments were conducted on a workstation with an Intel-10400, one NVIDIA 3090, and 64 GB RAM.

## G.2 Implementation Details

**TeCoS-LVM models**   For TeCoS-LVM models, all hyper-parameters on all datasets are fixed to be the same. We set the latent variable dimension to 32 and the hidden state dimension to 64 by default. We used the Adam optimizer ($\beta_1 = 0.9, \beta_2 = 0.999$) with a cosine-decay learning rate of 0.0003 with a mini-batch size 64. Training of these models is carried out for 64 epochs. We used the same architectures to implement all the TeCoS-LVM models (see Table 3). For LIF neuron TeCoS-LVM models (denoted as `TeCoS-LVM`), we used surrogate gradient learning (SGL) with an ERF surrogate gradient (see also Eq. 10) $\text{SG}_{\text{ERF}}(x) = \frac{1}{\sqrt{\pi}}\exp(-x^2)$. For Noisy LIF neuron TeCoS-LVM models (denoted as `TeCoS-LVM Noisy`), we used the Gaussian noise $\mathcal{N}(\epsilon; 0, 0.2^2)$ and the corresponding Noise-Driven Learning (a theoretically well-defined general form of SGL), which is described in

Appendix C, Eq. 15. Since random latent variables are involved in our model, we also employed the reparameterization trick for efficient training.

**Baselines**  We followed the settings in their original implementations for the CNN [9, 15] and IB-Disjoint [21] models. Some of these settings leverage the prior statistical structure information of the firing rate, thus improving the performance of these models [9]. In particular, the CNN model is trained with Gaussian noise injection, L-2 norm regularization (0.001) over the model parameters, and L1 norm regularization (0.001) over the predicted activations [9, 15]. The IB-Disjoint model is optimized with $\beta = 0.01$, which has proven to lead to better predictive power [21]. We also use the reparameterization trick for efficient training for the IB-Disjoint model. We use a constant learning rate of 0.001, a mini-batch size 64, and a default Adam optimizer for these two models. Following their original implementations, we used the early stopping technique in training. The network architectures of these models are listed in Appendix Table 3.

Table 3: List of network architectures (functional models) in our experiments. `conv` for the convolutional layer, `fc` for the fully-connected layer, `GRU` for the gated recurrent unit layer.

| MODEL NAME | DESCRIPTION |
|---|---|
| `TeCoS-LVM/TeCoS-LVM Noisy` (LIF/Noisy LIF spiking neurons, input channel=1); `TeCoS-LVM Rate` (LIF-Rate neurons input channel=1); `TeCoS-LVM Noisy Rate` (Noisy LIF-Rate neurons, input channel=1). | (feature extractor) 16`conv`25-32`conv`11-`fc`64 (real-valued RNN) GRU64 (encoder) `fc`64-`fc`64 (encoder mean) `fc`32 (encoder std) `fc`32 (prior) `fc`64-`fc`64 (prior mean) `fc`32 (prior std) `fc`32 (decoder) `fc`64-`fc`#RGCs |
| CNN (ReLU neurons, input channel=T) | 32`conv`25-BatchNorm 16`conv`11-BatchNorm `fc`#RGCs-BatchNorm ParametricSoftPlus |
| IB-Disjoint (ReLU neurons, input channel=T) | 16`conv`25-BatchNorm-32`conv`11-BatchNorm `fc`64-BatchNorm (encoder) `fc`64-BatchNorm-`fc`32-BatchNorm (encoder mean) `fc`32-BatchNorm (encoder std) `fc`32-BatchNorm (decoder) `fc`64-BatchNorm-`fc`#RGCs-BatchNorm ParametricSoftPlus |

Symbol descriptions (parameter `type` parameter): channel number `conv` kernel size; `fc` channel number; `GRU` hidden state dimension.

## H  Details of Ablation Experiments

### H.1  The effect of using spiking neurons

In this part, we constructed two variants (`TeCoS-LVM Rate` and `TeCoS-LVM Noisy Rate`) by replacing all the hidden spiking neurons in the TeCoS-LVM model with neurons that exhibit the same internal recurrence but provide non-spiking output. In other words, in these two variants, the activation of our hidden neurons transitioned from discrete spiking Heaviside functions to continuous functions. Specifically, we adopted the rate-output neuron model named GLIFR introduced in ref.[65]. In particular, the modified LIF-Rate neuron model we used here still uses the membrane update rules of LIF (and Noisy LIF). The output activation function of the LIF-Rate model is described by $o_t = \text{sigmoid}\left(\frac{u_t - v_{\text{th}}}{\sigma_u}\right)$, where the parameter $\sigma_u$ controls the smoothness of the membrane voltage-spike relationship. In doing so, the internal representation is constructed in a real-valued space rather than in a sparse spike space as TeCoS-LVM models with all LIF and Noisy LIF neurons.

