# OpenReview forum: "Temporal Conditioning Spiking Latent Variable Models of the Neural Response to Natural Visual Scenes"
_NeurIPS.cc/2023/Conference — NeurIPS 2023 poster_

### Official Review · Reviewer_qcrE · 2023-07-03

**Soundness:** 3 good
**Presentation:** 3 good
**Contribution:** 3 good
**Rating:** 6
**Confidence:** 3

**Summary:**

The authors propose a new model called temporal conditioning spiking latent variable models (TeCoSLVM) that uses spiking neurons to simulate neural response to visual stimuli. They claim that this approach helps to preserve information in spike trains and adapt to temporal dependencies. In experiments with retinal data, they demonstrate that the TeCoS-LVM models produce more realistic spike activities, fit spike statistics accurately, and generalize well to longer time scales.

**Strengths:**

Models that accurately predict spiking activity are still an open problem. The authors make an interesting contribution to this by defining a latent variable model that is trained with an information bottleneck objective. The work is, as far as I can tell, original. A few links to existing work are missing, which I mention below. The paper is mostly well and clearly written, and has an overall good quality. I have a few questions for the authors, but if they are answered satisfactorily, the paper makes a significant contribution to the development of dynamic spiking models.

**Weaknesses:**

I have a few questions regarding the paper and a few hints for relevant literature.
---
Decoder:
* I couldn't find information on the actual form of the decoder in the main paper. In particular, how is $\psi_{dec}$ computed?
* How are the latent states integrated into the LIF neurons?
* What is also not entirely clear to me is whether the model has access to real spike trains from previous time steps or not.
* From the figure it doesn't look like it, but since the decoder is not clear to me I cannot be sure. It would be nice if the authors could clarify. This is particularly relevant to make sure the comparison to the CNN is fair.

Experiments:
* How are the spiking rates computed?
* What loss function is used to train the CNN? Poisson loss? Would it be possible to also train the spiking model with Poisson loss (i.e. do you have the rate)? The reason I am asking that is that Fig 3B shows that your TECOS models perform better in terms of spike train dissimilarity. However they were trained on that. If the CNN was trained on a different objective, the comparison is a bit unfair.

Fig3:
* How do you generate spikes from the CNN in Fig 3A.

---
Regarding dynamic models of neural activity, these two paper might be relevant (in particular regarding the point of predicting variable length sequences):
- Fabian H. Sinz, Alexander S. Ecker, Paul G. Fahey, Edgar Y. Walker, Erick Cobos, Emmanouil Froudarakis, Dimitri Yatsenko, Xaq Pitkow, Jacob Reimer, Andreas S. Tolias Stimulus domain transfer in recurrent models for large scale cortical population prediction on video
- Eric Y Wang, Paul G. Fahey, Kayla Ponder, Zhuokun Ding, Andersen Change, Taliah Muhammad, Saumil Patel, Zhiwei Ding, Dat T. Tran, Jiakun Fu, Stelios Papadopoulos, Katrin Franke, Alexander S. Ecker, Jacob Reimer, Xaq Pitkow, Fabian H. Sinz, Andreas S. Tolias Towards a Foundation Model of the Mouse Visual Cortex

Regarding spiking neurons, this paper might be relevant:
- Ramesh, Poornima, Atayi, Mohamad, Macke, Jakob H: Adversarial training of neural encoding models on population spike trains

**Questions:**

I integrated the questions in the weaknesses above.

**Limitations:**

Limitations are discussed in one short paragraph at the end. It could be a bit more extensive.

---

> ### Author Rebuttal · Authors · 2023-08-09
>
> **Dear reviewer,**
>
> **Thanks very much for your detailed review and positive comments, which have greatly encouraged us! Our response is as follows.**
>
> > Decoder:
> >
> > 1. I couldn't find information on the actual form of the decoder in the main paper. In particular, how is $\psi_{dec}$  computed
> > 2. How are the latent states integrated into the LIF neurons
> > 3. What is also not entirely clear to me is whether the model has access to real spike trains from previous time steps or not
> > 4. From the figure it doesn't look like it, but since the decoder is not clear to me I cannot be sure. It would be nice if the authors could clarify. This is particularly relevant to make sure the comparison to the CNN is fair
>
> **Re:**  Thanks for your comment! We will improve relevant descriptions in our revised manuscript to make them clearer.
>
> 1. The temporal conditioning decoder $\psi\^{dec}$ is a two-layer spiking MLP that takes the current latent $\mathbf{z}\_t$ and hidden (which acts as a sensory memory) $\mathbf{h}\_{t-1}$ as inputs, and outputs neural response predictions.
> 2. Currently, in our implementation, we feed the latent states as the electric current input into the spiking neuron.
> 3. & 4.  The model will not access real spike trains (and stimuli) of previous timesteps (as described in Figure 1. C in the manuscript). Take time $t$ as an example, our model only takes the current stimulus $\mathbf{x}\_t$, and the sensory memory (hidden) $\mathbf{h}\_{t-1}$. This means, compared to the CNN model, our model receives fewer inputs (CNN receives a batch of stimuli [1,2], while we only have a single timestep input). Therefore, regarding the amount of input information received for producing a single-step prediction, our comparison is actually more favorable to CNN, but our model still performs better.
>
> [1] Deep learning models of the retinal response to natural scenes. *NeurIPS*, 2016.
>
> [2] Interpreting the retinal neural code for natural scenes ... *Neuron*, 2023.
>
> > Experiments:
> >
> > 1. How are the spiking rates computed
> > 2. * What loss function is used to train the CNN? Poisson loss? Would it be possible to also train the spiking model with Poisson loss (i.e. do you have the rate)
> >    * The reason I am asking that is that Fig 3B shows that your TECOS models perform better in terms of spike train dissimilarity. However they were trained on that. If the CNN was trained on a different objective, the comparison is a bit unfair.
>
> **Re:**
>
> 1. TeCoS models are spike-output, so in our tests, the firing rates are calculated using 20 repeated trials.
> 2. * Yes, the CNN model is trained using the Poisson loss [1]. Since our models directly output spike trains, it is not possible to *directly* use Poisson loss for them (additional conversions are required to achieve this, but the necessity of doing so is limited).
>    * Regarding the spike train distance comparison, we have conducted additional experiments that use different spike train distances to make to comparison more convincing (***Please refer to "Author Rebuttal by Authors" at the top of this page for TABLE 1 and 2.*** ). According to these results in Table 1, and Table 2, on these spike train distance metrics, our series of methods (tecos/tecos-noisy and our two variants) maintain a stable superiority over the baselines.
>
> [1] Deep learning models of the retinal response to natural scenes. *NeurIPS*, 2016.
>
> > Fig3: How do you generate spikes from the CNN
>
> **Re:** Following previous works [1,2], the spikes are generated by sampling from the predicted poisson distributions.
>
> [1] Deep learning models of the retinal response to natural scenes. *NeurIPS*, 2016.
>
> [2] Interpreting the retinal neural code for natural scenes... *Neuron*, 2023.
>
> > Regarding dynamic models of neural activity, these two papers might be relevant (in particular regarding the point of predicting variable length sequences):
> >
> > * Stimulus domain transfer in recurrent models for large scale ... *NeurIPS*, 2018.
> >
> > * Towards a Foundation Model of the Mouse Visual Cortex. *bioRxiv*, 2023
> >
> > Regarding spiking neurons, this paper might be relevant:
> >
> > * Adversarial training of neural encoding models ... *NeurIPS Workshop on NeuroAI*, 2019.
>
> **Re:** Thank you for pointing out the relevant literature!  We are very happy to add these relevant discussions to our revised manuscript to address your concern. We find that these excellent works (Sinz 2018, Wang 2023) leverage a hybrid "core" model (feedforward+recurrent) to shift along the temporal dimension. This is closely related to the temporal conditioning structure we used in this work. Regarding the related literature (Poornima 2019) you mentioned, we noticed that the part using adversarial training for optimization to capture the deterministic and stochastic components in neural population activities is very relevant to our motivation for using noisy LIF neurons.

---

> > ### Comment · Reviewer_qcrE · 2023-08-11
> > **Thanks for your clarifications**
> >
> > Dear authors, thanks for your clarifications. I have read the rebuttal. I will wait for the discussion with the other reviewers before adjusting (or keeping) my score.

---

> > > ### Author Response · Authors · 2023-08-18
> > > **Thanks for your response!**
> > >
> > > Thank you for your recognition of the idea of our work, as well as your efforts to improve our manuscript!
> > >
> > > We'll make sure that the points you raised are clear in the revised version of our manuscript.
> > >
> > > *Authors.*

---

### Official Review · Reviewer_QFPa · 2023-07-06

**Soundness:** 3 good
**Presentation:** 3 good
**Contribution:** 3 good
**Rating:** 7
**Confidence:** 3

**Summary:**

The authors model retinal ganglion cell responses to natural stimuli using a spiking latent variable model. They employ the (variational) Information Bottleneck (IB) method to compress the visual representation, similar to last year’s NeurIPS paper by Rahamni et al. However, this work differs in using binary responses (discretizes spike trains) instead of count responses, an input channel size of 1 instead of T, and temporal conditioning (RNN) instead of GP prior to handle temporal dependencies. They compare their results to Rahamni et al’s IB method and a CNN-based architecture on 4 salamander datasets. They show that those baselines fail to capture the spike autocorrelation, and a noisy version of their LVM model can also reproduce real firing rates more accurately.

**Strengths:**

- good performance
- scales well to long time series
- modeling single trial spike trains, not just trial-averaged firing rates

**Weaknesses:**

- Stimuli are converted to spikes, then to real valued signals (LVM), then back to spikes (ganglion cells). This does not align with visual processing in the retina: photoreceptors exhibit graded responses, as do bipolar cells (typically), whereas ganglion cells spike.
- The authors stress the importance of taking temporal dependencies into account, but instead of comparison to IB-GP they compare to IB-disjoint. Whereas the former has a temporal prior over latents the latter does not.
- A comparison to a method that also predicts spike trains not firing rates is lacking. Such a method could be competitive with regard to the considered spike train dissimilarity and autocorrelation measures.

**Questions:**

- Eq (1) 3rd eq with $u_t$ on both sides, yielding $u_t=u_{reset}/o_t$, is odd.

- line 103: Isn't $\tau_m$ is the membrane constant, not $\tau$? It seems $\tau=e^{-\Delta t/\tau_m}$ with bin size $\Delta t$.

- Why do you compare to IB-disjoint, not IB-GP? The authors of [19] favored IB-GP (temporal prior over latents) over IB-disjoint (no temporal prior).

- Does IB-GP need fewer latents that are more interpretable latents than yours?

- Line 290: Where is the Fig 5 you refer to?

- There are various measures of spike train dissimilarity in the literature, e.g. Victor-Purpura distance, Van Rossum distance, SPIKE-distance. You use the MMD in results and use an objective function (Eq 10) that specifically optimizes for this particular measure, thus obtaining good results. Do your conclusions hold for the other commonly used measures of spike train synchrony as well?


**Limitations:**

The authors have discussed limitations of their work.

---

> ### Author Rebuttal · Authors · 2023-08-09
>
> **Dear reviewer,**
>
> **Thank you very much for your thorough review and positive comments, which have greatly encouraged us! Our response is as follows.**
>
> > Stimuli are converted to spikes, then to real valued signals (LVM), then back to spikes (ganglion cells). This does not align with visual processing in the retina: photoreceptors exhibit graded responses, as do bipolar cells (typically), whereas ganglion cells spike.
>
> **Re:** Yes, the computational model reflects some, but not all, architectural, computational, and anatomical motifs of neural circuit formations [1-4].  The proposed dynamic spiking model aims to reproduce the neural activities by simulating the underlying general neural coding processes (as also mentioned in Reviewer qcrE); in doing so, it ignores some known neuronal biophysical motifs. And in this manuscript, we specialize our model to a natural stimuli case (as a paradigmatic example) for evaluations and investigations. We would like to add related discussions in the revised version of the manuscript to address your concern.
>
> [1] Stimulus-and goal-oriented frameworks ... *Nat. Neurosci.*, 2019.
>
> [2] The Tolman-Eichenbaum machine ... *Cell*, 2020.
>
> [3] Fitting summary statistics of neural data ... *NIPS*, 2021.
>
> [4] Mesoscopic modeling of hidden spiking neurons. *NIPS*, 2022.
>
> > Eq-1-3rd eq with $u_t$ on both sides, yielding $u_t=u_{reset}/o_t$ , is odd.
>
> **Re:**  Thanks for the correction. We will revise this expression in our revision to make it clearer. The original intention here is to indicate the resetting in the iterative calculation form we use, i.e., reset to $u\_{reset}$ after the neuron emits a spike.
>
> > line103: Isn't $\tau_m$ is the membrane constant, not $\tau$? It seems $\tau=\exp(-\Delta t / \tau_m)$ with bin size $\Delta t$.
>
> **Re:** Thanks for pointing out this unclear part. In our experiments, we actually handle data that is sampled using a fixed timestep. Thus, following previous work (e.g. [1]), when implementing spiking models in our case, we replace the genuine time constant with a fixed time constant which "incorporates"  $\Delta t$. That is, in our implementation in this case, the model weight is associated with the simulation timestep (although strictly, the model weights should be irrelevant to $\Delta t$). We will improve the presentations in our revision to make it clearer.
>
> [1] Brain-inspired global-local learning incorporated ... *Nat. Comm.*, 2022.
>
> > * The authors stress the importance of taking temporal dependencies into account, but instead of comparison to IBGP they compare to IB-disjoint. Whereas the former has a temporal prior over latents the latter does not. Why do you compare to IB-disjoint, not IBGP? The authors of [19] favored IBGP ...
> >
> > * Does IBGP need fewer latents that are more interpretable latents than yours?
>
> **Re:**
>
> * This is mainly because, in evaluations on our data, we found that the performances of IBDisjoint were better. E.g., on Mov1Ret1, the Pearson CC *(IBDisjoint vs IBGP)* 0.65vs 0.61. Actually, this phenomenon happens sometimes ("…IB-Disjoint obtains superior performance than IB-GP…" from Appendix 4 in [19]). Another possibility is that our re-implementations (as [19] has not yet released them) lack some technical details which are omitted in the paper [19].  Besides, IBGP has additional parameters (those of the Cauchy kernel) that need to be adjusted, this makes its hyper-parameter tuning process quite tricky when the data is changed.
> * In the main experiments in the IB paper, only nine neurons were actually used, so the latent dimension was small. Our tests show that our methods can actually, use a smaller latent dimension to produce more accurate predictions. For example, on Mov2Ret2, the firing rate Pearson CC of TeCoSLVM-Noisy with *latent-dim=8* was *0.81*, while those of the IB model with *latent-dim=32* was *0.66*. Thus, our method can use fewer latents, which is more advantageous from the interpretable aspect.
>
> > Line 290:Where is Fig5
>
> **Re:** Yes, in Appendix (it was re-uploaded as *Fig1* in the PDF file in "*Author Rebuttal by Authors*" panel). We will like to revise our manuscript to fix the unclear ref. you mentioned.
>
> > A comparison to a method that also predicts spike trains not firing rates is lacking...
>
> **Re:** To address your concern, we have conducted additional experiments using two variants of our model, following your comments (***Please refer to "Author Rebuttal by Authors" panel at the top of that page for TABLE 1, 2***). We noticed that, these variants also have certain advantages compared to the baseline, which also demonstrates the efficiency of our models.
>
> > There are various measures of spike train distances, e.g. Victor Purpura, van Rossum, SPIKE dist...
>
> **Re:** We have conducted additional evaluations on two benchmarks (Mov1Ret1, Mov2Ret2) following your comment and present them in tables as follows. According to these results (***Please refer to "Author Rebuttal by Authors" TABLE 1, and TABLE 2***), on these spike train distance metrics, our series of methods (tecos/tecos-noisy and two variants) maintain a stable advantage over the baselines.

---

> > ### Comment · Reviewer_QFPa · 2023-08-17
> >
> > Thank you for your clarifications. I appreciate the supplementary results for a model with real-valued hidden neurons, as well as the inclusion of additional evaluation metrics. These metrics further demonstrate that TeCoSLVM maintains its superiority over the baselines.
> > However, when considering these metrics, the noisy version of TeCoSLVM no longer holds the same preference. Could you provide clarification on this matter? Do you calculate these alternative metrics on individual trials, with subsequent averaging of the scores obtained, while for CC, trial averages are initially computed to derive rates before calculating the score?

---

> > > ### Author Response · Authors · 2023-08-18
> > >
> > > > Do you calculate these alternative metrics on individual trials, with subsequent averaging of the scores obtained, while for CC, trial averages are initially computed to derive rates before calculating the score?
> > >
> > > **Re:** Thank you for your reply. Yes, the spike train distance metrics are computed on multiple trials and then averaged; while for CC, trial averages are initially computed to get firing rates.
> > >
> > > > However, when considering these metrics, the noisy version of TeCoSLVM no longer holds the same preference.
> > >
> > > **Re:** It is true that when considering only the spike train distance metric, the performance of the noisy version of TeCoSLVM is slightly affected (as in *manuscript Fig. 3B, Mov1Ret2, Mov2Ret2*) due to the noise-perturbed neuronal dynamics. Although there is a slight decrease in spike timing precision, the overall impact of the noisy version is advantageous. Notably, the noisy version consistently and significantly improves the firing rate CC compared to the noise-free TeCoSLVM in all our evaluations. Another compelling merit is that, one can successfully and easily reproduce the trial-by-trial variance by incorporating these noisy spiking neurons.
> > >
> > >
> > >
> > > ---
> > >
> > > Finally, thanks for reviewing and improving our work. We'll ensure all points you raised are clear in our revision!
> > >
> > > *Authors*

---

### Official Review · Reviewer_qZ33 · 2023-07-08

**Soundness:** 2 fair
**Presentation:** 3 good
**Contribution:** 2 fair
**Rating:** 6
**Confidence:** 3

**Summary:**

This paper proposes a spiking latent variable model of neural response to natural visual stimuli. The model is trained to directly predict spike trains instead of trial-averaged firing rates, and designed to work, at test time, on sequences that are longer than sequences seen during training.

**Strengths:**

1. The authors identify two critical limitations of biophysically-realistic computational models. They propose a model that effectively address these limitations. Directly predicting spike trains using spiking neural networks, appears to be a natural solution.
2. The empirical experiments suggest that this results in a more accurate prediction of neural responses.

**Weaknesses:**

1. A main claim of the paper is that the model generalizes to longer time scales unseen during training. It is unclear whether the model is actually leveraging long-range interaction in its prediction. The results do not demonstrate that the memory of the model is holding long-range history from the sequence. It is unclear whether a model with a sliding window (of 1s) would perform the same or worse.
2. While the authors do a great job at analyzing different aspects of the model including the impact of hyperparameters on the learned latent space, it remains unclear how this more bio-realistic model provides novel insights into sensory processing.

**Questions:**

- The model can be applied to sensory data of other modalities, how sensitive is the model to the choice of hyperparameters? For example, the latent dimension or the hidden dimension?
- Why was the warmup period similar for all four datasets (0.5s), one would expect to observe differences due to different dynamics of the stimuli.

**Limitations:**

As stated, the proposed model is only partially realistic.

---

> ### Author Rebuttal · Authors · 2023-08-09
>
> **Dear reviewer, many thanks for your detailed review and comments! Our response is as follows.**
>
> > A main claim of the paper is that the model generalizes to longer time scales [...]
>
> **Re:**  We would like to point out that our model only uses single-step stimulus input for prediction and leverages temporal dependencies by holding a memory in its predictions (as mentioned by Reviewer *QFPa*, *qcrE*). Also, the IB-Disjoint model has an LVM structure and uses a 1s sliding window, similar to the model you described. According to experimental results in Fig.2 and 3 in the paper, our model outperform IB-Disjoint in all evaluations.
>
> And to further address your concern, we implemented a 1s sliding window variant (*SlidWindVariant*, and *SlidWindVariant Noisy*) of our model following your description, and ran tests on Mov1Ret1 data (Table below). The results show that these sliding window variants performed worse in all metrics, reflecting the advantages of using a temporal conditioning structure to handle these temporal interactions.
>
> | metric\model       | TeCoSLVM | TeCoSLVMNoisy | SlidWindVariant | SlidWindVariant Noisy | CNN    | IBDisj. |
> | ------------------ | -------- | ------------- | --------------- | --------------------- | ------ | ------- |
> | firing rate CC     | 0.579    | 0.727         | 0.567           | 0.681                 | 0.690  | 0.653   |
> | VictorPurpuraDist. | 12.84    | 14.02         | 18.62           | 19.93                 | 19.60  | 21.92   |
> | vanRossumDist.     | 127.35   | 238.61        | 307.78          | 343.02                | 376.82 | 394.02  |
> | SPIKE Dist.        | 0.124    | 0.155         | 0.197           | 0.210                 | 0.207  | 0.224   |
>
> > While the authors do a great job at analyzing different aspects of the model including [...], it remains unclear how this more bio-realistic model provides novel insights into sensory processing.
>
> **Re:**  To name some concrete sensory processing (encoding of natural stimuli) studies that could benefit from our model. Fitting neural response to natural stimuli more accurately: works like [1] could benefit from our model to identify microcircuits that mediate/gate diverse encoding properties in sensory processing. Intrinsic/latent dynamics, specific circuit mechanism characterization:  our model provides an efficient approach to decompose the computations in sensory processing into the latents (a generative model that enables a low-dimension representation [2]), which could allow neuroscientists to generate new hypotheses, or test their hypotheses about how interneurons with diverse response properties are combined to perform sensory processing [2-4] and provide neurally-grounded understandings of sensory computations therein.
>
> [1] Multiplexed computations... *Nat. Comm.*, 2017
>
> [2] Stimulus-and goal-oriented frameworks... *Nat. Neurosci.*, 2019.
>
> [3] Toward a unified theory of efficient... *PNAS*, 2018.
>
> [4] Interpreting the retinal neural... *Neuron*, 2023.
>
> > The model can be applied to sensory data of other modalities, how sensitive is the model to the choice of hyperparameters? For example, the latent dimension or the hidden dimension?
>
> **Re:** Our results show that the model is quite robust to the hyperparameters you mentioned and can achieve good performance even when the hidden state/latent space dimension is small.
>
> We evaluated the performance of TeCoS-LVM models under different hidden state and latent space dimensionality settings.  The results, shown in Appendix Fig. 6A, indicate that increasing the latent space dimension improved performance. Still, further increases ($>32$)  had little effect when the latent space dimension was large. This also suggests that the number of latent factors for the visual stimuli coding task we currently consider is not very large. On the other hand, increasing the hidden state dimension enhances sensory memory capacity and benefits temporal conditioning operations. As shown in Appendix Fig. 6B, the performance does not increase significantly when hidden state dimensionality increases to a certain extent (about $64$). The results in Appendix Fig. 6B also indicate that the proposed temporal conditioning mechanism can effectively utilize sensory memory and achieve good results even when the hidden state dimension is small (memory capacity is small).
>
> > Why was the warmup period similar for all four datasets (0.5s), one would expect to observe differences due to different dynamics of the stimuli.
>
> **Re:**  Thanks for your meticulous review. In fact, the setting of this warmup period is currently, more empirical. This means, *we cannot use it as an "indicator" to reflect the different dynamics of different data*. The length of this period is more empirical, as it is difficult to precisely choose the most "suitable" warmup period length for different data. In other words, currently, we lack a reliable quantitative approach to accurately measure which warmup length is exactly "optimal" or most "correct".

---

> > ### Comment · Reviewer_qZ33 · 2023-08-19
> > **Thanks for your clarifications**
> >
> > Thank you for addressing the points I raised! These clarifications and the additional work carried out in response to all reviewers further strengthens this paper. I increased my score.

---

> > > ### Author Response · Authors · 2023-08-20
> > > **Thanks for your response**
> > >
> > > Dear reviewer,
> > >
> > > We'd like to express our appreciation for your review and insightful suggestions. Your feedback will greatly improve our work. We'll ensure all points you raised are clear in our revised manuscript.
> > >
> > > *Authors*

---

### Official Review · Reviewer_j9bD · 2023-07-09

**Soundness:** 3 good
**Presentation:** 2 fair
**Contribution:** 2 fair
**Rating:** 5
**Confidence:** 2

**Summary:**

This paper proposes a model, which is composed of spiking neural networks and a conventional recurrent network, to reproduce neuronal responses, i.e., retinal ganglion cells. The author claims that the novelty of this model is incorporating the spiking network, which makes the model directly outputs spikes and is more biologically plausible. The author did simulations to show the model outperforms alternatives in reproducing retinal ganglion cells' activities.

**Strengths:**

### Originality
In my understanding, the paper has two novelties: 1) the model incorporates spiking neural networks as encoder and decoder; 2) the model uses an RNN to provide temporal conditioning which is more flexible than previous models whose kernels have fixed temporal durations.

### Quality
The writing has clear explanations of the probabilistic framework, but it is quite unclear how the probabilistic framework is connected with underlying spiking networks (see my comments below).

**Weaknesses:**

### Major
I understand the variational information bottleneck framework, and the text of this part is clearly written. However, it is unclear the details of the spiking networks and how they are incorporated into the probabilistic framework.
1) Specifically, how the distributions' parameters $\phi$ and $\psi$ in Eqs. 6 and 7 are related to spiking networks? PS: I only see line 151 saying $\phi$ is a linear readout of spiking neurons' responses.
2) What are the structures of spiking networks in the encoder and decoder? Are the pure feedforward spiking network, or recurrent spiking network?
3) During the model training, did the authors only train the readout weight from the spiking network? or also train the synaptic weights inside the spiking network?

Without providing these details, I am not confident in judging the novelty of this paper. I am happy to see the authors' rebuttal to explain this.

### Minor
- Line 177, “Sec. C” and Line 188, “Sec. E”: I don’t know where these sections are. In the supplementary?
- Eq. 3: does this equation miss a term of $\beta I_c$ in the Lagrangian multiplier?

**Questions:**

- Conceptually, I don't understand why the model with spiking network encoder and decoder could render better results. I am really interested in some deep, theoretical discussions on this issue. I could understand if the output layer is a spiking network, it would fit the spiking neuronal responses better. Nonetheless, I don't understand why using spiking net as intermediate layers could improve the performance. Can the author provide an alternative result to replace the spiking intermediate layers by rate-based neurons and compare the performance of the two versions?

- Another conceptual question. From the neural pathway point of view, the neural circuits from the retina to the visual thalamus, i.e., LGN, is equivalent to dimensionality reduction because the number of neurons decreases with the neural pathway. And then from LGN to the visual cortex the neuron number will expand a lot. In this sense, the visual pathway inside the retina, and the one from the retina to LGN looks like an __encoder__, while the pathway from LGN to the visual cortex is like a __decoder__. If my statement was true, the retina is just an encoder with less number of neurons in the output layer (retinal ganglion cells) compared with the input layer (photoreceptor). Hence, to reproduce retinal ganglion cells' responses, what is the advance of considering the proposed model architecture (Fig. 1B) with an encoder (dimensionality reduction) and a decoder (dimensionality expansion?

**Limitations:**

See my comments in Questions.

---

> ### Author Rebuttal · Authors · 2023-08-09
>
> **Dear reviewer, thank you very much for your detailed review and positive comments, which have greatly encouraged us. Our response is as follows.**
>
> > I understand the variational information bottleneck framework, and the text of this part is clearly written. However, it is unclear the details of the spiking networks and how they are incorporated into the probabilistic framework.
> >
> > 1. Specifically, how the distributions' parameters $\phi$ and $\psi$ in Eqs. 6 and 7 are related to spiking networks?
> > 2. What are the structures of spiking networks in the encoder and decoder? Are the pure feedforward [...]
> > 3. During the model training, did the authors only train the readout weight from the spiking network [...]
>
> **Re:**
>
> 1. For the temporal conditioning prior $\phi\^{prior}$ , we use a spiking network, which takes the hidden state $\mathbf{h}\_t$ as input, and outputs the distribution means and variances using linear readout synapses. The temporal conditioning encoder $\psi\^{enc}$ is a spiking network that receives the current stimulus $\mathbf{x}\_t$ and hidden $\mathbf{h}\_{t-1}$ as inputs, and produce mean and variance using linear readout synapses. The temporal conditioning decoder $\psi\^{dec}$ takes the latent $\mathbf{z}\_t$ and the hidden $\mathbf{h}\_{t-1}$ as inputs, and produces spike activities (predictions).
>
> 2. The spiking networks (for parameterizing those $\phi$ and $\psi$s) mentioned above are all feed-forward; they are spiking MLPs. Since we introduced temporal conditioning operations using hidden states to handle temporal dependencies, we followed the feed-forward network structure, as in previous works [1,2]. We found that such a simple feed-forward structure can already produce very competitive performance. Adding more recurrent connections in the model may bring some improvement, but it also increases the complexity and the number of parameters of the model, which is not conducive to further analysis. Therefore, in this work, we only consider using a feed-forward structure.
> 3. During the model training, we optimize all synaptic weights inside the model, including those of the spiking networks, and the RNN, which maintains the hidden $\mathbf{h}$.
>
> [1] Deep learning models of the retinal response to natural scenes. *NIPS*, 2016.
>
> [2] Interpreting the retinal neural code for natural scenes ... *Neuron*, 2023.
>
> > Line 177, “Sec. C” and Line 188, “Sec. E”: I don’t know where these sections are. In the supplementary?
> >
> > Eq. 3: does this equation miss a term of $\beta I_c$ in the Lagrangian multiplier?
>
> **Re:**  Yes, they are in the Appendix. We'd like to revise our presentation in revision to make it clearer.  The $I_c$ term is not missing in equation 3, because we are maximizing an equivalent objective [1].
>
> [1] Deep Variational IB. *ICLR*, 2016.
>
> > Conceptually, I don't understand why the model with spiking network encoder and decoder could render better results [...]
>
> **Re:** We have conducted additional experiments following your comments (***please refer to "Author Rebuttal by Authors" panel above for Table 1,2***). We additionally consider *Victor-Purpura*, *van-Rossum*, and *SPIKE* distances as metrics. The results show that our variants can also achieve competitive performance than baselines, but lag behind standard tecos-lvm models.
>
> Since our main network is feed-forward, using spiking neurons can enhance the capture of the spatiotemporal features embedded in the stimuli stream (which is essentially spatiotemporal) through the neuronal temporal dynamics (ANN neurons don't have that), thus producing better predictions. Also, a hallmark of natural stimuli is their sparse latent structure [1,2]. By using spiking neurons in the intermediate layer, the model can take advantage of the sparse representation (compared with those of ANN neurons) to better fit the latent structure of natural stimuli, resulting in improved performances.
>
> [1] *Natural image statistics...* Springer, 2009.
>
> [2] Toward a uniﬁed theory of efﬁcient, predictive... *PNAS*, 2018
>
> > Another conceptual question. From the neural pathway point of view, the neural circuits from the retina to the visual thalamus, i.e., LGN, is equivalent to dimensionality reduction because the number of neurons decreases with the neural pathway. And then from LGN to the visual cortex the neuron number will expand a lot. In this sense, the visual pathway inside the retina, and the one from the retina to LGN looks like an **encoder**, while the pathway from LGN to the visual cortex is like a **decoder**. [...] Hence, to reproduce retinal ganglion cells' responses, what is the advance of considering the proposed model architecture (Fig. 1B) with an encoder (dimensionality reduction) and a decoder (dimensionality expansion?
>
> **Re:**  Considering the proposed architecture allows us to simultaneously obtain latent dynamics for future interpretation/analyses [1-3] and achieve better modeling performance.
>
> The encoder-decoder we use here is not solely for dimensionality expansion/reduction (Fig.1A is only schematic), but rather to simulate a two-stage efficient neural coding process [4] of inferring latent codes from stimuli and then decoding neural responses from latent codes. And, the latent dimension in our model is adjustable, e.g., if we choose a large latent dimension (larger than the output RGC dim.), then the whole process becomes dimension reduction.  So the entire TECOS-LVM model is performing the encoding process (stimuli to the retina), which is, functionally, an encoder as you mentioned. We would like to supplement the relevant content in the revised manuscript and improve the graphics in Fig. 1 to address your concern.
>
> [1] Inferring single-trial neural population dynamics... *Nat. Methods*, 2018.
>
> [2] Targeted neural dynamical modeling. *NIPS*, 2021.
>
> [3] Drop, swap, and generate... *NIPS*, 2021.
>
> [4] Toward a uniﬁed theory of efﬁcient, predictive... *PNAS*, 2018

---

> > ### Comment · Reviewer_j9bD · 2023-08-18
> >
> > Thanks for the authors' reply, which addresses most of my questions.
> >
> > I am still unclear about which rate-model was used in your additional experiments. Is it the CNN in the table?
> > The perceptrons in CNN don't have temporal dynamics. A fair comparison would be just replacing the spike generation in spiking neurons with a transfer function to output instantaneous rate. PS: I do agree the temporal dynamics of spiking neurons can capture the spatiotemporal features in the stimuli, but I don't get the point of why the spikes can help the performance of the model.

---

> > > ### Author Response · Authors · 2023-08-20
> > > **Thanks for your reply!**
> > >
> > > > Thanks for the authors' reply, which addresses most of my questions.
> > > >
> > > > I am still unclear about which rate-model was used in your additional experiments. Is it the CNN in the table?
> > > >
> > > > The perceptrons in CNN don't have temporal dynamics. A fair comparison would be just replacing the spike generation in spiking neurons with a transfer function to output instantaneous rate. PS: I do agree the temporal dynamics of spiking neurons can capture the spatiotemporal features in the stimuli, but I don't get the point of why the spikes can help the performance of the model.
> > >
> > > **Re:**  Yes,  in the table of our additional experiments, the CNN and the IBDisjoint are rate-model.
> > >
> > >
> > >
> > > Firstly, incorporating spiking neurons in intermediate layers allows the model to take advantage of the sparse spike representation to better fit the sparse latent structure of natural stimuli [1], which might enhance the model performance. On the other hand, we employed a spike train-based objective rather than a firing rate-based one. In this scenario, spiking hidden neurons can better align with the ultimate objective, which may also contribute to improved performance.
> > >
> > >
> > >
> > > Due to the limited time of the discussion period, we regret that these supplementary experiments about using the "Leaky-Integrate and Firing Rate" you mentioned are still in progress. However, we are delighted to include those experiments and discussions in our subsequent revision to address your concern.
> > >
> > > [1] *Natural image statistics... * Springer, 2009.
> > >
> > > ------
> > >
> > > Thank you very much for reviewing and providing constructive suggestions. They have greatly helped us enhance the quality of our manuscript. We'll make improvements in our revised manuscript accordingly.
> > >
> > > *Authors*

---

### Author Rebuttal · Authors · 2023-08-09

We sincerely thank all *Reviewers* for their thorough reviews and *Chairs* for their efforts. We have responded to each point in your comments separately. *Due to word count limitations, we had to shorten some of your reviews when quoting them.* Please find our replies in your corresponding panels.

-----

Below, we refer to the two variants (from Rev. **j9bD** and **QFPa**) of the proposed tecos-lvm as the model ***"variant"*** (which has the same structure as tecos-lvm, only uses LIF output neurons, other hidden neurons are real-valued), and the model ***"variant Noisy"*** ( same as the ***variant***, but uses noisy LIF output neurons). Results are means computed across repeats. Also, we added Victor Purpura, van Rossum, and SPIKE distances (the lower the better) as evaluation metrics (from Rev. **QFPa** and **qcrE**).



***TABLE 1:*** Comparison with variants on Mov1Ret1. CC stands for Pearson Correlation Coefficients (the higher the better). The TeCoS-LVM, and TeCoS-LVM Noisy models are the main models introduced in our manuscript.

| metric \ model | TeCoSLVM | TeCoSLVM Noisy | variant | variantNoisy | CNN    | IBDisjoint |
| -------------- | -------- | -------------- | ------- | ------------ | ------ | ---------- |
| firing rate CC | 0.579    | 0.727          | 0.468   | 0.720        | 0.690  | 0.653      |
| Victor Purpura | 12.84    | 14.02          | 14.58   | 13.11        | 19.60  | 21.92      |
| van Rossum     | 127.35   | 238.61         | 179.12  | 245.45       | 376.82 | 394.02     |
| SPIKE          | 0.124    | 0.155          | 0.144   | 0.136        | 0.207  | 0.224      |

***TABLE 2:*** Comparison with variants on Mov2Ret2. CC stands for Pearson Correlation Coefficients (the higher the better). The TeCoS-LVM, and TeCoS-LVM Noisy models are the main models introduced in our manuscript.

| metric \ model | TeCoSLVM | TeCoSLVM Noisy | variant | variantNoisy | CNN     | IBDisjoint |
| -------------- | -------- | -------------- | ------- | ------------ | ------- | ---------- |
| firing rate CC | 0.616    | 0.822          | 0.586   | 0.798        | 0.708   | 0.656      |
| Victor Purpura | 22.67    | 28.44          | 23.71   | 28.49        | 39.26   | 38.38      |
| van Rossum     | 574.30   | 1135.81        | 682.51  | 1205.68      | 2638.96 | 2244.98    |
| SPIKE          | 0.123    | 0.153          | 0.133   | 0.148        | 0.221   | 0.221      |

----

---

### Decision · Program_Chairs · 2023-09-21

**Decision:**

Accept (poster)

**Comment:**

The reviewers were unanimous in their appraisal of this paper's contributions as above the bar for acceptance to NeurIPS.  I congratulate the authors on their detailed rebuttals, which were critical in leading some reviewers to raise their scores.  I'm pleased to report that this paper has been accepted to NIPS.  Congratulations!  Please revise the manuscript to address all reviewer comments and questions.